# TSSC4 is a component of U5 snRNP that promotes tri-snRNP formation

Klára Klimešová [1,6], Jitka Vojáčková [1,6], Nenad Radivojević [1], Franck Vandermoere [2],
Edouard Bertrand [3,4,5], Celine Verheggen [3,4,5] & David Staněk [1✉]

U5 snRNP is a complex particle essential for RNA splicing. U5 snRNPs undergo intricate biogenesis that ensures that only a fully mature particle assembles into a splicing competent U4/U6•U5 tri-snRNP and enters the splicing reaction. During splicing, U5 snRNP is substantially rearranged and leaves as a U5/PRPF19 post-splicing particle, which requires regeneration before the next round of splicing. Here, we show that a previously uncharacterized protein TSSC4 is a component of U5 snRNP that promotes tri-snRNP formation. We provide evidence that TSSC4 associates with U5 snRNP chaperones, U5 snRNP and the U5/PRPF19 particle. Specifically, TSSC4 interacts with U5-specific proteins PRPF8, EFTUD2 and SNRNP200. We also identified TSSC4 domains critical for the interaction with U5 snRNP and the PRPF19 complex, as well as for TSSC4 function in tri-snRNP assembly. TSSC4 emerges as a specific chaperone that acts in U5 snRNP de novo biogenesis as well as post-splicing recycling.

[1] Institute of Molecular Genetics, Czech Academy of Sciences, Prague, Czech Republic. [2] Institut de Génomique Fonctionnelle, University of Montpellier, CNRS, INSERM, Montpellier, France. [3] Institut de Génétique Moléculaire de Montpellier, University of Montpellier, CNRS, Montpellier, France. [4] Equipe labélisée Ligue Nationale Contre le Cancer, Montpellier, France. [5] Present address: Institut de Génétique Humaine, Montpellier, France. [6] These authors contributed equally: Klára Klimešová, Jitka Vojáčková. ✉email: stanek@img.cas.cz

Pre-mRNA splicing is a key step in protein expression catalyzed by one of the largest complexes in the cell called the spliceosome. The human spliceosome is a dynamic multi-component molecular machine that assembles de novo on each intron to be removed and needs to be recycled after each round of splicing. Over 160 proteins and 5 small noncoding nuclear RNAs (snRNAs) are involved in spliceosome formation and rearrangements during RNA splicing[1]. SnRNAs together with associated proteins form small nuclear ribonucleoprotein particles (snRNPs), which are basic building blocks of the spliceosome. Almost all human introns are removed by the major snRNPs named U1, U2, U4, U5, and U6, while a small fraction of introns (U12 type) is removed by the minor spliceosome consisting of U11, U12, U4atac, and U6atac. U5 snRNP is shared by both major and minor pathways (reviewed in ref. [2]).

U5 snRNP contains U5 snRNA tightly associated with a heptameric ring of Sm proteins and eight U5-specific proteins. Sm ring assembly on U5 snRNA is promoted and controlled by SMN and PRMT5 complexes in the cytoplasm and the core snRNP (U5 snRNA+7 Sm proteins) is imported to the nucleus where it interacts with additional U5 proteins[3–7]. Four of the U5-specific proteins PRPF8/U5-220K, SNRNP200/hBrr2/U5-200K, EFTUD2/hSnu114/U5-116K, and SNRNP40/WDR57/U5-40K form a stable RNA-free Heterotetrameric Complex (RHC)[8]. Yeast and human PRPF8 associate with the chaperone AAR2, which helps to properly fold PRPF8[9–11]. PRPF8 then interacts with EFTUD2 and SNRNP200[12]. In yeast, Aar2p is exchanged with Brr2p[13,14], while in humans AAR2 co-purifies with all four RHC members[12], which indicates that AAR2 stays associated until the whole RHC is formed. Initial phases of RHC formation are further promoted and controlled by the R2TP/HSP90 chaperone, which interacts both directly and indirectly via a bridging factor ZNHIT2 with EFTUD2[12,15,16]. PRPF8 and likely the whole pre-formed RHC then interacts with the core U5 snRNP in nuclear Cajal bodies[7] and this step is promoted by another chaperone ECD[17]. Finally, the last four U5-specific proteins PRPF6/U5-102K, DDX23/U5-100K, CD2BP2/U5-52K, and TXNL4A/hDib1/Dim1/U5-15K join the complex, but little is known about this step of U5 snRNP biogenesis. Three of these proteins, namely PRPF6, CD2BP2, and TXNL4A directly interact with each other[18–20] but whether they need the assistance of any chaperone to be incorporated into maturing U5 snRNP is unknown.

Before U5 snRNP can participate in splicing it needs to associate with U4/U6 di-snRNP to create the splicing competent U4/U6•U5 tri-snRNP. During tri-snRNP assembly, CD2BP2 protein is released and at least three additional proteins are added[18,21,22]. The PRPF6 protein is the key factor that interacts with U5 and U4/U6 proteins and bridges both particles in the tri-snRNP[19,23–25]. Consistently, downregulation of PRPF6 inhibits the tri-snRNP formation and results in sequestration of individual tri-snRNP components in Cajal bodies[24,26].

During splicing reaction, the whole spliceosome and particularly the tri-snRNP undergo significant rearrangements. All di-snRNP proteins and a half of U5-specific proteins leave during spliceosome activation and only RHC and Sm proteins remain bound to U5 snRNA[1]. When splicing is completed, a post-splicing complex is released by the concerted action of RNA helicases DHX8/PRPF22 and DHX15/PRPF43[27–30]. The post-splicing U5 snRNP leaves the spliceosome together with the PRPF19 complex as a 35S particle[31,32]. How U5 snRNP is recycled from this complex into the active tri-snRNP form is currently unknown.

U5 chaperones AAR2 and ZNHIT2 co-precipitate with U5-specific proteins and several poorly characterized factors, including ECD, TSSC4, NCDN[12,15]. Among these proteins, only ECD has been shown to interact with U5 snRNP and its chaperones and promote U5 snRNP assembly[15,17,33,34]. Here, we focused on the protein TSSC4, which repeatedly co-purified with U5-specific proteins as well as with AAR2 and ZNHIT2 chaperones[12,15,35] and analyzed its potential role in snRNP metabolism. Human TSSC4 has been identified as a gene that escapes imprinting in an otherwise imprinted genomic region[36]. A single-nucleotide polymorphism in TSSC4 has been associated with reduced body height in humans but the molecular function of this protein is unknown[37]. In this work, we provide evidence that TSSC4 associates with the U5 snRNP and the U5/PRPF19 complex. We identified TSSC4 domains important for interactions with the U5 snRNP and the PRPF19 complex and show how TSSC4 downregulation affects tri-snRNP formation. Based on our results, we propose that TSSC4 is a key protein involved in both de novo biogenesis as well as post-splicing recycling of U5 snRNP.

## Results

**TSSC4 associates with mono-U5 and U5/PRPF19 complexes.** To get insight into a function of TSSC4 in snRNP biogenesis, we first mapped its interaction partners by an unbiased quantitative proteomic approach based on stable isotope labeling in cell culture in conjunction with immunoprecipitation (SILAC IPs)[12,38,39]. We tagged TSSC4 with either GFP or FLAG, expressed labeled proteins in HeLa cells, and immunopurified them with anti-GFP or anti-FLAG antibodies, respectively (Fig. 1a and Supplementary Data 1). The proteomic analysis of both GFP and FLAG IPs revealed a strong association of TSSC4 with Sm proteins and all U5-specific proteins including CD2BP2, a marker of mono-U5 snRNP, which is not a component of the U4/U6•U5 tri-snRNP[18]. We also identified several U5 chaperones such as AAR2, ZNHIT2, RUVBL1, RUVBL2, and ECD and a putative chaperone EAPP that has been shown to interact with U5 proteins PRPF8 and EFTUD2[12,15,17]. In addition, tagged TSSC4 co-purified with low levels of PRPF19, a core component of the PRPF19 complex. All detected TSSC4 interactors are listed in Supplementary Data 1.

To confirm the SILAC-IP results, we immunoprecipitated TSSC4-GFP and TSSC4-FLAG and detected co-precipitated proteins by western blotting (Fig. 1b and Supplementary Fig. 1a). This result confirmed that TSSC4 interacts with U5 snRNP and the PRPF19 complex and co-precipitated all U5-specific proteins, Sm proteins, and core subunits of the PRPF19 complex (PRPF19 and CDC5L). At the same time, TSSC4 did not pull down SART3, a marker of U4/U6 di-snRNP, the component of U4/U6 and U4/U6•U5 particles PRPF31 and U2-specific SNRPA1 (U2A'). Consistently with the set of co-precipitated proteins, TSSC4-GFP also pulled down specifically the U5 snRNA (Fig. 1c). We did not detect any TSSC4 self-interaction because ectopically expressed TSSC4-FLAG did not pull down endogenous TSSC4 (Supplementary Fig. 1b). We noticed that TSSC4 migrates in SDS-polyacrylamide gels around 50 kDa while its predicted molecular weight is 34 kDa (Supplementary Fig. 1b). TSSC4 is an acidic protein with the isoelectric point 4.81. A similar shift in apparent molecular weight has been described for another acidic U5-specific protein CD2BP2 with isoelectric point 4.49[18]. Finally, we provided evidence that all the detected interactions of TSSC4 with U5 and PRPF19 complex proteins are RNA independent (Fig. 1d, see also control RNA gel at Supplementary Fig. 1c). These data show that TSSC4 associates specifically with U5 snRNP-containing complexes and that this binding is based on protein–protein interactions.

To further characterize TSSC4 containing complexes, we immunoprecipitated TSSC4-FLAG, eluted precipitated complexes with a FLAG peptide, and resolved them by ultracentrifugation in a glycerol gradient (Fig. 2a, bottom panel). We observed two

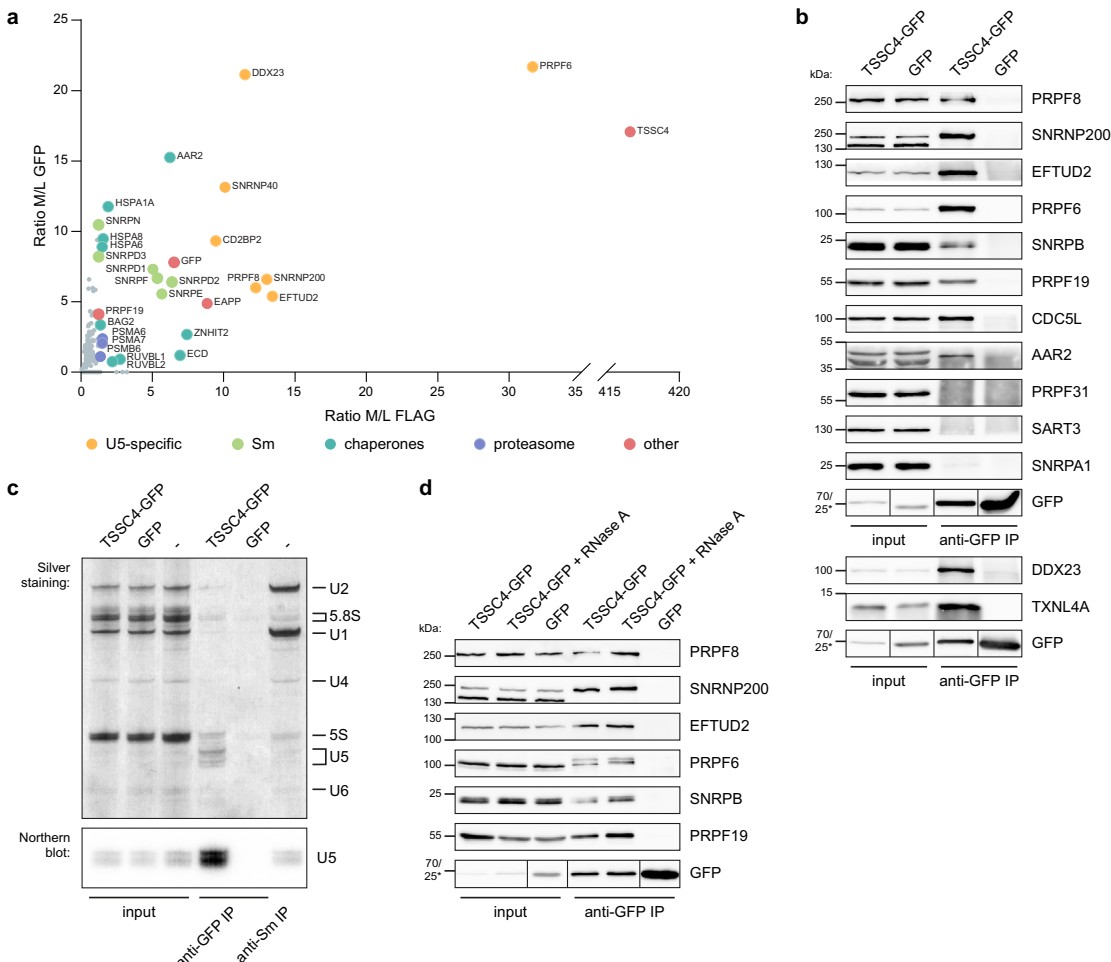

**Fig. 1 TSSC4 associates specifically with U5 snRNP and PRPF19 protein. a** Characterization of proteins co-precipitating with TSSC4-GFP and TSSC4-FLAG by SILAC-IP. Primarily U5-specific proteins (yellow), Sm proteins (green), chaperones involved in U5 biogenesis (blue) co-purified with TSSC4. The ratio M/L shows a SILAC ratio of medium labeled proteins co-precipitated with tagged TSSC4 (M) and control immunoprecipitation from unlabeled cells (L). The experiment was performed once with TSSC4-GFP and once with TSSC4-FLAG. **b** TSSC4 interacts with proteins specific for U5 snRNP and the PRPF19 complex. TSSC4-GFP immunoprecipitation followed by western blotting and detection of proteins specific for U5 (PRPF8, SNRNP200, EFTUD2, and PRPF6—top panel and DDX23 and TXNL4A—bottom panel), U2 (SNRPA1) and U4/U6 (PRPF31 and SART3) snRNPs and PRPF19 complex (PRPF19 and CDC5L). SNRPB is one of the Sm proteins common to all snRNPs, and AAR2 is a PRPF8 chaperone. The positions of molecular weight markers are shown, the marker labeled by asterisk applies only to the GFP sample in the bottom combined GFP panels. See also Supplementary Fig. 1 for TSSC4-FLAG immunoprecipitation. **c** TSSC4 precipitates specifically U5 snRNA. TSSC4-GFP immunoprecipitation followed by RNA isolation and detection of co-purified RNAs by silver staining (top panel) and Northern blotting with a probe against U5 snRNA (bottom panel). Immunoprecipitation of Sm proteins with Y12 antibody served as a positive control and GFP as a negative control. **d** TSSC4 association with U5-specific and PRPF19 proteins is RNA independent. TSSC4-GFP immunoprecipitation from RNase A treated or untreated cell lysates was followed by detection of U5-specific proteins and PRPF19. The positions of molecular weight markers are shown, the marker labeled by asterisk applies only to the GFP sample in the bottom combined GFP panel. See Supplementary Fig. 1c for control of the RNase treatment. **b**, **c** The experiments were performed three times or **d** two times with similar results. **b**–**d** Uncropped and unprocessed scans are provided in the Source Data file.

discrete U5 snRNA-containing complexes. One complex sedimented in fractions 14–16, which was a position where the mono-U5 snRNP migrated when the nuclear extract was resolved on an identical gradient run in parallel (Fig. 2a, top panel). Consistent with the prediction that this complex represents the U5 snRNP, we detected U5-specific proteins PRPF8 and EFTUD2 but not a U4/U6 di-snRNP marker PRPF4 in these fractions (Fig. 2b). We also observed endogenous TSSC4 (marked by an asterisk) sedimenting in nuclear extracts in the same fractions as the U5 snRNP. Surprisingly, we detected only traces of TSSC4-FLAG (marked by double asterisks) co-sedimenting with U5 snRNP, which indicates that the FLAG tag might partially destabilize TSSC4 association with U5 snRNP. A second U5 snRNA-containing complex precipitated by TSSC4-FLAG sedimented in

lower fractions 19–20 where we detected U5-specific proteins PRPF8 and EFTUD2 and PRPF19. These data indicate that TSSC4 interacts with two distinct U5 snRNA-containing particles, one can be identified as a mono-U5 snRNP and the second complex contains U5 and PRPF19 complex markers, which is consistent with a spectrum of TSSC4 associated proteins detected by immunoprecipitations (Fig. 1).

**U5 snRNP and the PRPF19 complex interact with different TSSC4 domains.** Next, we decided to determine TSSC4 protein domains important for interactions with U5 snRNP and the PRPF19 complex. Analysis of the human TSSC4 sequence by BLAST and Phyre2 did not reveal any significant homology to

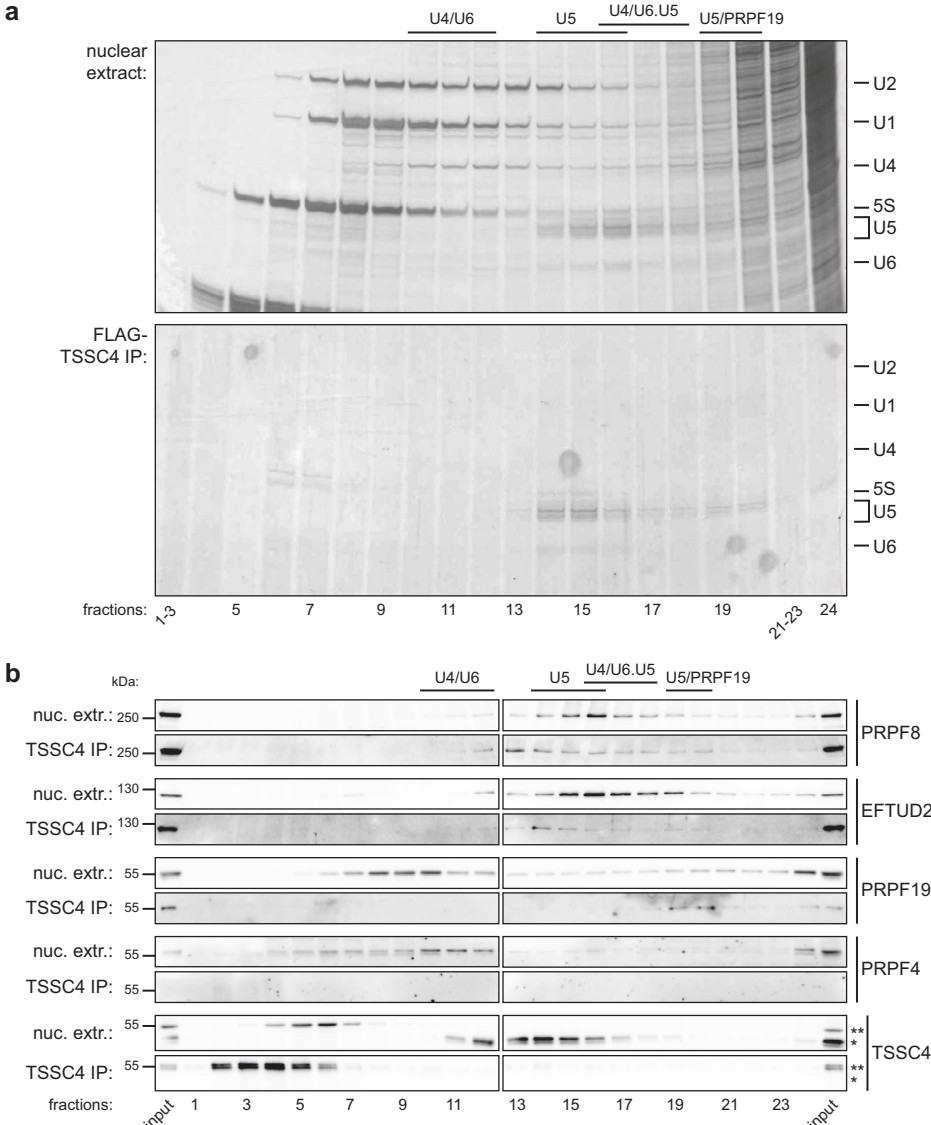

**Fig. 2 TSSC4 interacts with two U5 snRNA-containing complexes. a** Complexes co-precipitating with TSSC4-FLAG were eluted by a FLAG peptide and resolved by ultracentrifugation on 10–30% glycerol gradient. RNA was isolated from each fraction, resolved on UREA-PAGE gels, and silver-stained (bottom). A whole-nuclear extract was run in parallel to identify fractions where individual snRNPs sediment (top). **b** After RNA preparation (**a**), proteins were isolated from the organic phase and detected by western blotting. Lighter complex sedimenting in fractions 14–16 contains primarily U5 snRNA and U5-specific proteins (PRPF8 and EFTUD2) and heavier complex migrating in fractions 19–20 contains U5 snRNA, U5-specific proteins, and the PRPF19 protein. Endogenous TSSC4 marked by an asterisk (*), TSSC4-FLAG marked by double asterisks (**). The positions of molecular weight markers are shown. **a**, **b** The experiment was performed two times with a similar result. Uncropped and unprocessed scans are provided in the Source Data file.

known protein domains. We, therefore, decided to perform an unbiased screen and deleted consecutive windows of 50 amino acids (aa) from TSSC4 protein (Fig. 3a). All deletion mutants were tagged with GFP at the C-terminus and transiently expressed in HeLa cells. All TSSC4 constructs localized primarily to the cell nucleus and only the Δ201–250 construct localized weakly also to the cytoplasm (Supplementary Fig. 2a). TSSC4-GFP full-length and deletion mutants were immunoprecipitated by anti-GFP antibodies and co-precipitation of U5 snRNP-specific proteins and all four core PRPF19 complex subunits (PRPF19, CDC5L, PLRG1, and BCAS2) was monitored by western blotting (Fig. 3b). In parallel, we also monitored co-purified RNAs (Supplementary Fig. 2b). We identified two protein regions that were important for TSSC4 interaction with its partners. Deletion of aa 51–100 (Δ51–100) reduced TSSC4 interaction with U5 snRNP components (Fig. 3b, top panel and Supplementary

Fig. 2b) and removal of aa 201–250 (Δ201–250) almost completely abolished association with the PRPF19 complex (Fig. 3b, bottom panel). We also noticed a differential effect of the 51–100 aa deletion on the interaction with U5 subunits. While U5 snRNA, SNRPB/SmB, CD2BP2, and PRPF6 binding was completely lost upon 51–100 aa deletion, only partial reduction was observed in the case of RHC proteins PRPF8, SNRNP200, and EFTUD2. Unexpectedly, the interaction of the Δ51–100 mutant with SNRNP40, which is also a component of RHC was also abolished. In addition, a small but reproducible reduction of SNRNP200 and PRPF8 binding was also observed in the case of the Δ201–250 deletion mutant. Deletion of the last 30 aa at the C-terminus destabilized the protein. We, therefore, expressed and immunoprecipitated the last 50 C-terminal aa tagged with GFP but did not detect any interaction with U5 snRNP or PRPF19 complex components.

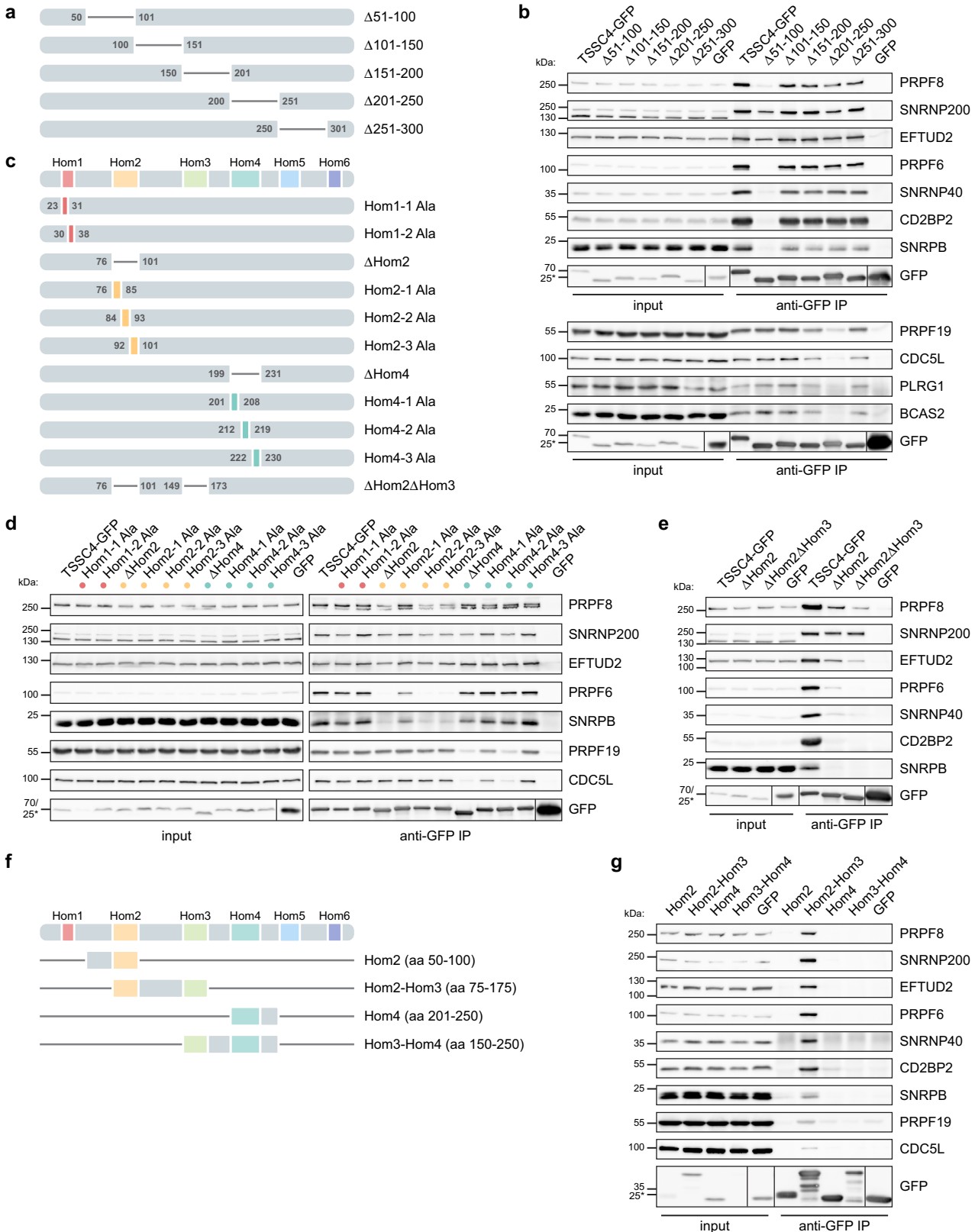

A protein sequence comparison of TSSC4 done using T-Coffee tool[40] across the animal kingdom revealed six conserved domains, which we named Hom1–Hom6 (Supplementary Fig. 2c and Fig. 3c). These homology regions largely correspond to conserved regions identified previously by comparing human and mouse TSSC4 sequences[41]. We also performed an in silico analysis of TSSC4 ordered/disordered regions (Supplementary Fig. 2d) using MetaDisorderMD2[42,43]. This analysis predicted that most TSSC4 sequence is largely disordered with a few exceptions involving the central part of the molecule spanning Hom3 and 4 domains, which shows a weak tendency for ordered structures.

**Fig. 3 TSSC4 contains two conserved domains interacting independently with U5 snRNP and the PRPF19 complex. a** A schematic representation of TSSC4 deletion mutants. **b** Co-immunoprecipitation of U5 and PRPF19 complex markers with TSSC4 deletion mutants. Deletion of aa 51–100 abolished interaction with several U5-specific proteins (PRPF6, SNRNP40, CD2BP2, and SNRPB). The same deletion reduced the association of TSSC4 with U5 proteins PRPF8, SNRNP200, and EFTUD2. Deletion of aa 201–250 specifically inhibited association with PRPF19 complex markers (bottom panel). See also Supplementary Fig. 2a, b for TSSC4 mutant localization and co-precipitation of snRNAs. **c** A schematic representation of TSSC4 homology regions and deletion/substitution mutants. See also Supplementary Fig. 2c for TSSC4 homology alignment. **d** Co-immunoprecipitation of U5 and PRPF19 complex markers with TSSC4 deletion and substitution mutants. Deletion or mutation of the Hom2 region reduced interaction with U5-specific proteins and deletion/mutation of the Hom4 region inhibited association with the PRPF19 complex. See also Supplementary Fig. 2e for TSSC4 mutant localization. **e** Simultaneous deletion of conserved domains Hom2 and Hom3 further reduced association of TSSC4 with U5 proteins PRPF8 and EFTUD2. The experiment was performed only once. See also Supplementary Fig. 2e for ΔHom2ΔHom3 mutant localization. **f** A schematic representation of TSSC4 fragments. **g** Co-immunoprecipitation of U5 and PRPF19 complex markers with TSSC4 fragments. Expression of the central part of TSSC4 containing Hom2 and Hom3 regions is sufficient for interaction with U5-specific proteins and a weak association with the PRPF19 complex. See also Supplementary Fig. 2f for localization of these constructs. **b**, **d**, **e**, **g** The positions of molecular weight markers are shown, the markers labeled by asterisk apply only to the GFP samples in the bottom combined GFP panels. Uncropped and unprocessed scans are provided in the Source Data file. **b**, **d**, **g** The experiments were performed three times with similar results.

The 51–100 aa window overlaps with the conserved Hom2 region while aa 201–250 contains the conserved Hom4 sequence. To further narrow down interaction domains of TSSC4 with U5 snRNP and PRPF19 complex, we prepared several additional constructs (Fig. 3c). First, we deleted 24 aa (V77–L100) that overlap Δ51–100 and Hom2 regions (ΔHom2 construct) and 31 aa (S200–R230) that intersect Δ201–250 and Hom4 sequences (ΔHom4 construct). We also mutated eight or six conserved aa in Hom2 and Hom4 domains into alanines. All constructs containing GFP at the C-terminus were expressed transiently in HeLa cells. They localized into the cell nucleus and weakly also in the cytoplasm in the case of the ΔHom4 and Hom4-3_Ala constructs (Supplementary Fig. 2e). All TSSC4-GFP constructs were immunoprecipitated using anti-GFP antibodies and co-precipitated components of U5 and PRPF19 complexes were monitored by western blotting (Fig. 3d). Deletion of the conserved regions Hom2 and Hom4 had the same effect as deletion of aa 51–100 and 201–250, respectively. Removal of Hom2 region reduced the interaction with U5 snRNP, and deletion of Hom4 reduced association with the PRPF19 complex. More detailed mutagenesis revealed 16 conserved aa (M85–L100) essential for interaction with PRPF6 and SNRPB proteins (Hom2-2_Ala and Hom2-3_Ala; Fig. 3d). Substitution of these aa with alanines also partially reduced but not completely abolished association with U5 proteins PRPF8, SNRNP200, and EFTUD2. We further identified six conserved aa in the middle of the Hom4 domain (V213-P218) that are essential for association with the PRPF19 complex (Hom4-2_Ala). We also removed Hom1 region close to the N-terminus but this deletion destabilized the whole protein. We, therefore, substituted Hom1 aa into alanines (Hom1-1_Ala and Hom1-2_Ala). We still observed reduced stability of Hom1-1_Ala but Hom1-2_Ala was expressed at similar levels as other TSSC4 constructs (see GFP signal in inputs at Fig. 3d). We observed a weak phenotype on SNRNP200 binding in the case of Hom1-1_Ala construct, which might reflect a lower expression of this construct. A similarly weak but reproducible reduction of the TSSC4-SNRNP200 association was observed for ΔHom2 and Hom2-2_Ala, Hom2-3_Ala substitution mutants and for ΔHom4 and Hom4-2_Ala constructs.

Deletion of the central part of TSSC4 (aa151–200) did not reveal any binding phenotype despite it contained the most conserved region of TSSC4–Hom3. We, therefore, created a double ΔHom2ΔHom3 mutant, tagged it with GFP and transiently expressed in HeLa cells. The double mutant also localized into the cell nucleus (Supplementary Fig. 2e). Analysis of U5 subunits co-precipitating with the double mutant revealed a significant reduction of all U5 components and only SNRNP200 binding was unchanged (Fig. 3e).

To identify a minimal part of TSSC4 that recapitulates the binding profile of the full-length protein, we created several constructs spanning various regions of the central part of TSSC4 whose deletion reduced interaction with U5 snRNP and PRPF19 complex (Fig. 3f). All constructs were tagged with GFP and after transient expression in HeLa cells immunoprecipitated with anti-GFP antibodies (Fig. 3g). Hom2 alone did not associate with any of the tested proteins but together with Hom3, this construct pulled down all tested U5-specific and Sm proteins. In addition, we also detected a weak association with PRPF19 complex components PRPF19 and CDC5L, which likely reflect immuno-precipitation of the U5/PRPF19 complex. All constructs localize equally to the nucleus and the cytoplasm (Supplementary Fig. 2f) and TSSC4 fragments containing domain 2 (Hom2) and domains 3 and 4 (Hom3-Hom4) were partially destabilized and expressed on a lower level than other constructs (Fig. 3g, see GFP signal).

These results show that three conserved regions bind different partners. Hom2 is essential and together with Hom3 also sufficient for association with U5 snRNP. Both domains make specific contacts with U5 subcomplexes where Hom3 primarily interacts with PRPF8 and EFTUD2 proteins that form a subcomplex early during biogenesis[12] and Hom2 mediates interaction with the mature U5 particle. Hom4 domain is essential but not sufficient for binding of the PRPF19 complex. We did not identify a mutant that would completely abolish TSSC4 association with SNRNP200, which indicates multiple contacts between these two proteins. To further narrow TSSC4 interaction partners, we tested TSSC4 interaction with all U5-specific proteins and the PRPF19 protein in yeast two-hybrid assay. However, we did not detect any positive interaction with any of the tested proteins (Supplementary Table 1). Given these negative results and the fact that we found multiple interaction platforms for U5 snRNP, we speculate that TSSC4 makes multiple weak contacts with several U5 snRNP and PRPF19 complex proteins.

**TSSC4 can contact U5-specific proteins already in the cytoplasm.** The interaction between TSSC4 and AAR2 suggests that TSSC4 can associate with U5 proteins early during biogenesis. To test whether TSSC4 can interact with U5-specific proteins already in the cytoplasm, we performed a tethering assay that we previously applied to monitor interactions of chaperones AAR2 and R2TP with PRPF8[12]. We fused TSSC4 with a fragment of DDX6/p54 that targets the fusion protein to P-bodies, and with dsRed to monitor the fusion protein localization[44]. We expressed the fusion protein TSSC4-DDX6-dsRed in HeLa cells together with five U5-specific proteins and SNRPD1/SmD1 (Fig. 4). We observed strong P-body localization of PRPF8-GFP and weaker

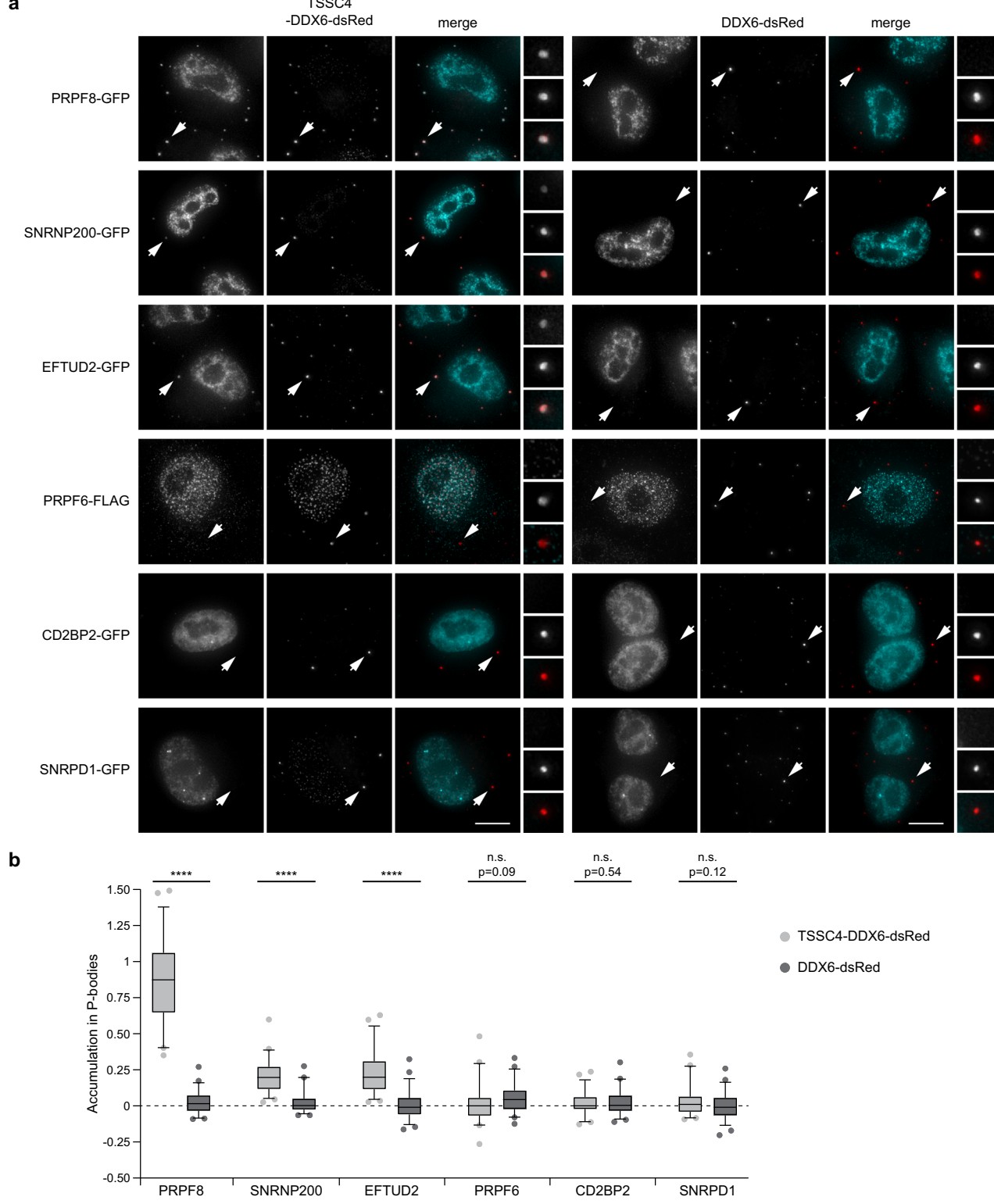

**Fig. 4 TSSC4 interacts with PRPF8, SNRNP200, and EFTUD2 in the cytoplasm. a** HeLa cells co-transfected with TSSC4-DDX6-dsRed fusion protein (red) and U5-specific proteins tagged with GFP or FLAG (turquoise). The DDX6/p54 peptide targets the TSSC4-DDX6-dsRed protein into cytoplasmic P-bodies. Co-recruitment of individual U5 proteins (PRPF8, SNRNP200, and EFTUD2) into P-bodies indicates their interaction with TSSC4. P-bodies marked by arrows are enlarged three times in insets. Scale bar represents 10 μm. **b** Quantification of P-body localization. The signal of GFP/FLAG-tagged U5-specific proteins in P-bodies was corrected by subtraction of the signal in the surrounding cytoplasm and compared to the signal in the nucleus. In total, 50 P-bodies were analyzed for each condition and shown in a box blot (center line—median; box limits—1st and 3rd quartiles; whiskers—5th and 95th percentile), two lowest and two highest values are depicted as separate dots. Statistical significance was determined by the two-tailed *T* test, and all *P* values lower than 0.0001 are indicated by ****. Original values for box plots are provided in the Source Data file.

but also clear tethering of EFTUD2-GFP and SNRNP200-GFP. However, these proteins make a preassembled particle already in the cytoplasm[12] and we therefore cannot unambiguously determine which of these proteins is the binding partner for TSSC4. U5-specific proteins PRPF6-FLAG and CD2BP2-GFP, and SNRNPD1-GFP were not accumulated in P-bodies when co-expressed with the TSSC4-DDX6-dsRed fusion protein. These data suggest that TSSC4 interacts preferentially with RHC components, and this interaction can occur already in the cytoplasm before their import to the nucleus. Based on these results and the fact that TSSC4 associates with known RHC chaperones AAR2, ZNHIT2, and RUVBL1/2 as well as all U5-specific proteins and U5 snRNA (Fig. 1 and ref. [12]), we conclude that TSSC4 associates with RHC subunits during their biogenesis in the cytoplasm and stays associated till the complete U5 snRNP is formed.

However, TSSC4 is mainly nuclear protein and this model of action predicts that it is constantly exported to the cytoplasm and reimported to the nucleus. To test whether TSSC4 shuttle between the nucleus and the cytoplasm, we performed a heterokaryon shuttling assay. TSSC4-GFP and PRPF8-GFP as negative control were expressed in HeLa cells together with mCherry, which freely diffuses between the nucleus and the cytoplasm and serves as a marker of successful fusion. After proteosynthesis inhibition with cycloheximide or puromycin, HeLa cells were fused with mouse NIH-3T3 fibroblasts and the presence of GFP and mCherry signals in mouse nuclei monitored after 3 h (Supplementary Fig. 3). While PRPF8-GFP remained in HeLa nuclei, TSSC4-GFP clearly localized to mouse nuclei showing its ability to shuttle between the nucleus and the cytoplasm.

**TSSC4 promotes U4/U6•U5 tri-snRNP formation.** Based on the TSSC4 interaction profile we speculated that TSSC4 could play a role in U5 snRNP metabolism. We and others have previously shown that the accumulation of U4, U5, and U6 snRNAs in Cajal bodies represents a convenient readout of the U5 snRNP assembly and recycling status. For example, inhibition of U5 snRNP maturation by PRPF8 or R2TP downregulation results in the accumulation of U5, U4, and U6 snRNAs in Cajal bodies[7,12]. In contrast, inhibition of post-spliceosomal complex recycling caused specific accumulation of U4 and U6 snRNAs but not U5 snRNA[45]. To test a function of TSSC4 in U5 snRNP biogenesis and recycling, we reduced TSSC4 expression by RNA interference and assayed localization of snRNAs in Cajal bodies (Fig. 5). TSSC4 downregulation increased Cajal body accumulation of U4, U5, and U6 snRNAs but not U2 snRNA, which indicated specific defects in U5 and/or tri-snRNP biogenesis. To learn more about the identity of U5 snRNP sequestered in the Cajal body, we analyzed the presence of several U5-specific proteins in Cajal bodies after TSSC4 knockdown. Surprisingly, the Cajal body localization of all tested U5-specific proteins was also increased after TSSC4 knockdown (Fig. 6). In addition, TSSC4 knockdown induced Cajal body accumulation of mono-U5-specific marker CD2BP2, which indicated that the mono-U5 snRNP localized to Cajal bodies. In contrast, we did not observe increased Cajal body localization of PRPF19, which suggested that the U5/PRPF19 complex did not accumulate in Cajal bodies. These data indicated that in the absence of TSSC4, U5 snRNP accumulates in Cajal bodies but this U5 particle is not able to make a contact with U4/U6 di-snRNP and form the functional tri-snRNP, resulting in the accumulation of U4, U5, and U6 snRNAs in this nuclear structure.

This was a rather unexpected finding because inhibition of tri-snRNP assembly by PRPF31 and PRPF6 knockdown induced Cajal body accumulation of U4 and U6 snRNAs only[24]. We therefore first wanted to confirm that U5 snRNP proteins and specifically

PRPF6, which is the U5-specific protein that interacts with U4/U6 di-snRNP and is essential for tri-snRNP assembly[23,46], are present in U5 snRNP after TSSC4 downregulation. TSSC4 was downregulated in HeLa cells stably expressing PRPF8-GFP[47], PRPF8-GFP was immunoprecipitated using anti-GFP antibodies and co-precipitation of PRPF6 and other U5-specific proteins assayed by western blotting (Fig. 7a). We did not notice any difference in protein composition of U5 snRNP in control cells and cells treated with anti-TSSC4 siRNA, which showed that the removal of TSSC4 did not significantly affect the composition of the U5 snRNP and in particular the tri-snRNP bridging factor PRPF6 was still present.

To confirm that TSSC4 is important for U4/U6•U5 tri-snRNP formation, we knocked down TSSC4 and analyzed snRNP complexes by ultracentrifugation in glycerol gradients (Fig. 7b). TSSC4 downregulation resulted in redistribution of U5 snRNP proteins from fractions where U4/U6•U5 tri-snRNP migrates into lighter fractions, indicating inhibition of tri-snRNP formation or instability of the tri-snRNP particle. These data are consistent with fluorescence results (Figs. 5–6) and together show that TSSC4 is important for tri-snRNP assembly and/or stability. If TSSC4 depletion reduces tri-snRNP concentration then it should also affect RNA splicing. To test this prediction, we downregulated TSSC4 by RNAi and analyzed the splicing efficiency of seven genes (Fig. 7c). Downregulation of PRPF8, which was used as a positive control, reduced splicing in six cases. TSSC4 reduction exhibited a much milder effect and we observed significantly reduced splicing of two genes (*LDHA* and *RPL19*). It should be noted that PRPF8 knockdown also partially destabilized TSSC4 but TSSC4 downregulation did not alter PRPF8 expression (Supplementary Fig. 4).

Finally, we decided to probe whether TSSC4 domains essential for binding of U5 snRNP (Hom2) and the PRPF19 complex (Hom4) are also important for tri-snRNP biogenesis. We downregulated endogenous TSSC4 by RNAi, expressed GFP tagged wild-type TSSC4 or deletion/substitution mutants resistant to anti-TSSC4 siRNA, and monitored U5 snRNA localization in Cajal bodies as a readout of tri-snRNP biogenesis (Fig. 8a, b). Expression of the full-length TSSC4 clearly reduced U5 snRNA accumulation in Cajal bodies, indicating rescue of the tri-snRNP assembly and the specificity of the knockdown. This rescue was mostly independent of the TSSC4-GFP expression level (Supplementary Fig. 5). TSSC4 lacking the Hom2 domain partially rescued the phenotype but was about 2 times less efficient than wild-type TSSC4. Expression of constructs where PRPF19 interacting domain was missing (ΔHom4) or mutated (Hom4-2_Ala) significantly reduced U5 snRNA accumulation in Cajal bodies but the rescue was also less efficient than in the case of the full-length TSSC4. These data indicate that both the U5 snRNP interaction domain Hom2 and the PRPF19 binding domain Hom4 are important for the correct function of TSSC4 in tri-snRNP assembly.

## Discussion

In this manuscript, we describe a function for the previously uncharacterized protein TSSC4. We show that TSSC4 interacts (Fig. 1) and co-sediments (Fig. 2) with the U5 snRNP but we did not find any indication that TSSC4 is a stable component of the U4/U6•U5 tri-snRNP. Based on these data, we propose that TSSC4 is a mono-U5 snRNP-specific protein. TSSC4 also co-precipitates known U5 snRNP chaperones AAR2, ZNHIT2, ECD, and R2TP, which specifically function during assembly of RHC[9–12,15,33]. This finding suggests that TSSC4 is involved in specific steps of U5 snRNP maturation involving RHC. This conclusion is consistent with TSSC4 ability to shuttle between the nucleus and the cytoplasm (Supplementary Fig. 3), and our

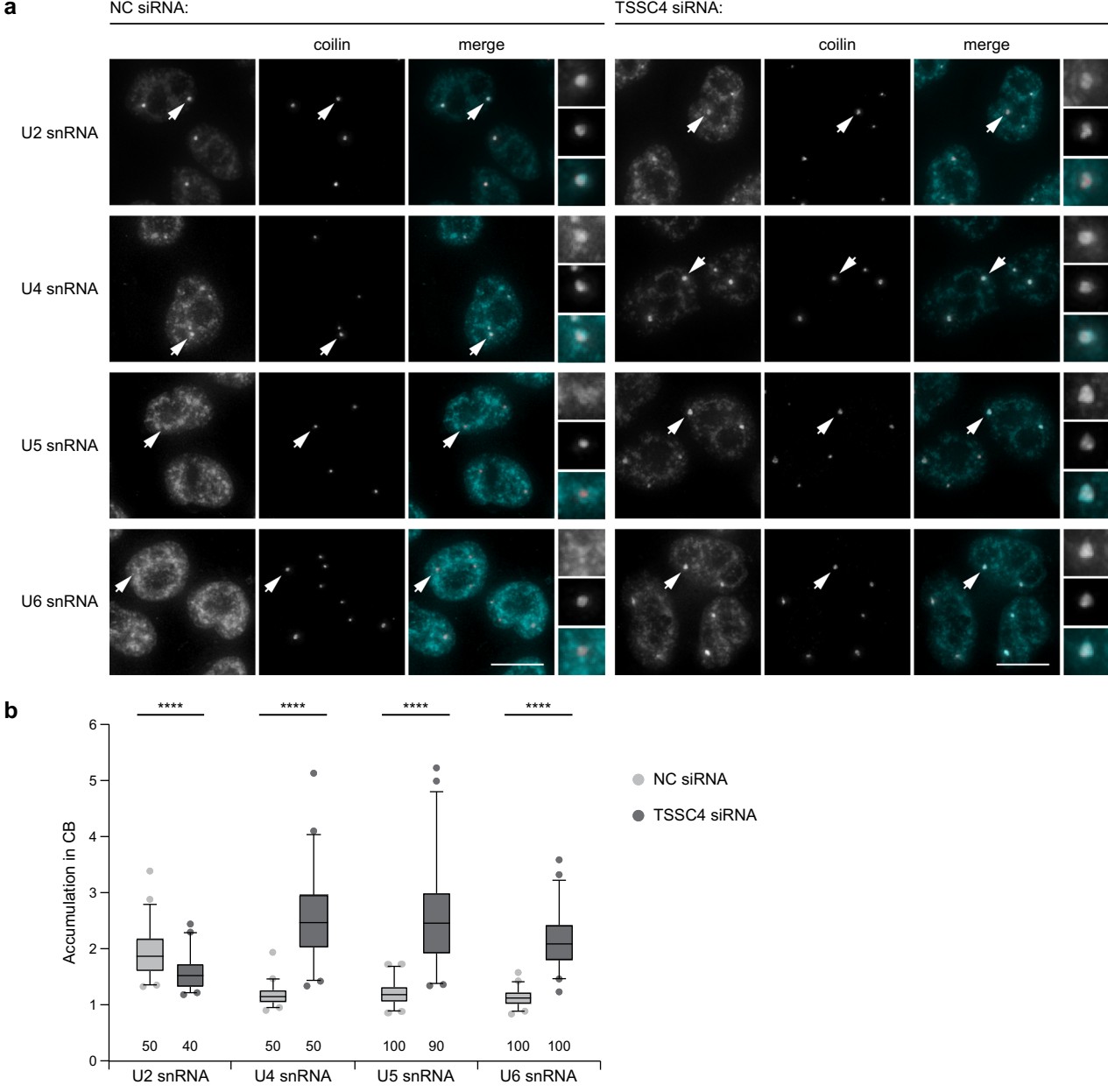

**Fig. 5 TSSC4 downregulation increases the accumulation of U4, U5, and U6 snRNAs in Cajal bodies. a** Knockdown of TSSC4 by RNAi increased the accumulation of U4, U5, and U6 snRNAs in Cajal bodies. snRNAs detected by in situ hybridization (turquoise), Cajal bodies visualized by immunodetection of their marker coilin (red). Cajal bodies marked by arrows are enlarged 3 times in insets. Scale bars represent 10 μm. **b** Quantification of Cajal body accumulation. The fluorescence signal of in situ hybridization probes in Cajal bodies was normalized to signal in the nucleoplasm and shown in a box plot (center line—median; box limits—1st and 3rd quartiles; whiskers—5th and 95th percentile or 2.5th and 97.5th percentile for U2, U4 snRNAs, or U5, U6 snRNAs, respectively). Two lowest and two highest values are depicted as separate dots in each box plot. The number of analyzed cells is shown at the bottom of the diagram. Data for U2 and U4 snRNAs were measured in one experiment, data for U5 and U6 snRNAs were obtained in two biologically independent experiments. Statistical significance was determined by the two-tailed T test, and all P values lower than 0.0001 are indicated by ****. Original values for box plots are provided in the Source Data file.

tethering experiment revealing that TSSC4 is able to interact with RHC components PRPF8, SNRNP200, and EFTUD2 already in the cytoplasm where these proteins form a stable complex (Fig. 4)[12]. However, TSSC4 downregulation did not reveal any defects in U5 snRNP composition (Fig. 7), and all the tested mono-U5-specific proteins accumulated in Cajal bodies after TSSC4 knockdown (Fig. 6). These results strongly indicate that TSSC4 is not important for the incorporation of individual U5 snRNP components into a maturing particle. However, TSSC4 downregulation caused defects in tri-snRNP formation

and/or stability (Figs. 5–7). Based on the fact that TSSC4 is not a stable component of the tri-snRNP and thus it is difficult to imagine how it can stabilize this particle, we propose that TSSC4 promotes tri-snRNP assembly. Because all components are present in the U5 snRNP upon TSSC4 knockdown, we propose that TSSC4 functions as a chaperone that facilitates U5 snRNP association with the U4/U6 di-snRNP.

RHC and Sm proteins remain associated with U5 snRNA after splicing while other U5 and tri-snRNP-specific proteins dissociate[48]. In addition, PRPF19 and IBC/XAB2 complexes and

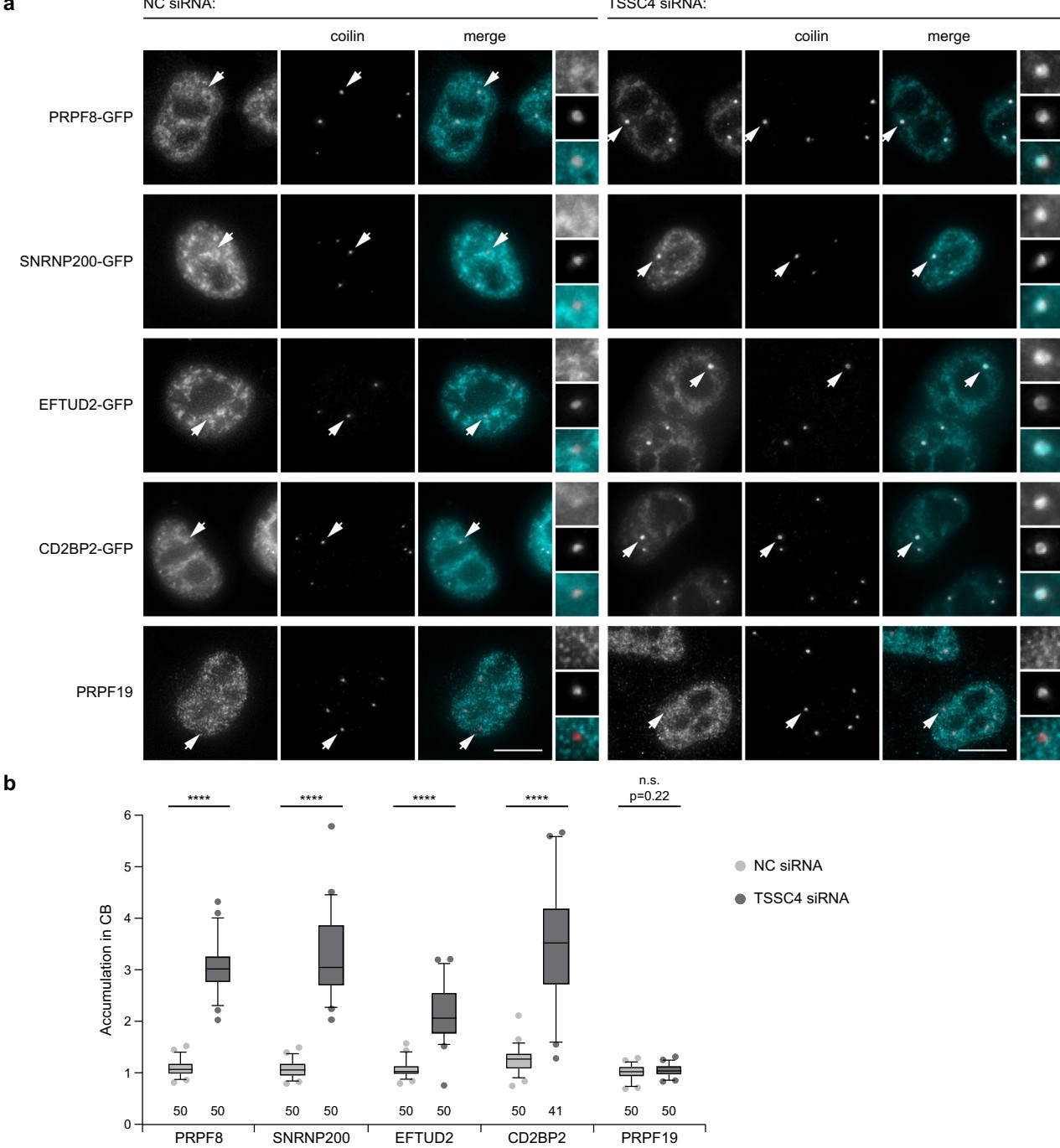

**Fig. 6 TSSC4 downregulation increases the accumulation of U5-specific proteins in Cajal bodies. a** Knockdown of TSSC4 by RNAi increased the accumulation of U5-specific proteins in Cajal bodies. PRPF19 was visualized by indirect immunofluorescence, the other proteins were tagged with GFP and stably (PRPF8-GFP and SNRNP200-GFP) or transiently (CD2BP2-GFP and EFTUD2-GFP) expressed in HeLa cells (turquoise). Cajal bodies were visualized by immunodetection of coilin (red). Cajal bodies marked by arrows are enlarged three times in insets. Scale bars represent 10 μm. **b** Quantification of Cajal body accumulation. Fluorescence signal in Cajal bodies was normalized to signal in the nucleoplasm and shown in a box plot (center line—median; box limits—1st and 3rd quartiles; whiskers—5th and 95th percentile). Two lowest and two highest values are depicted as separate dots in each box plot. The number of analyzed cells is shown at the bottom of the diagram. Statistical significance was determined by the two-tailed $T$ test, and all $P$ values lower than 0.0001 are indicated by ****. Original values for box plots are provided in the Source Data file. **a**, **b** Two biologically independent experiments were performed for each condition and combined.

their associated factors form together with U5 snRNP the post-splicing 35S particle[31,32]. Because TSSC4 interacts with all core components of the PRPF19 complex (Figs. 1–3) and a complex containing U5 snRNA and PRPF19 co-purifies with TSSC4 (Fig. 2), we conclude that TSSC4 associates with the post-splicing

U5/PRPF19 particle. Interestingly, a single-nucleotide polymorphism in the PRPF19 complex-interacting Hom4 region of TSSC4 has been associated with lower human height[37], suggesting that this phenotype might be due to a defect in U5 snRNP recycling. Our co-purification assays suggest several modes of

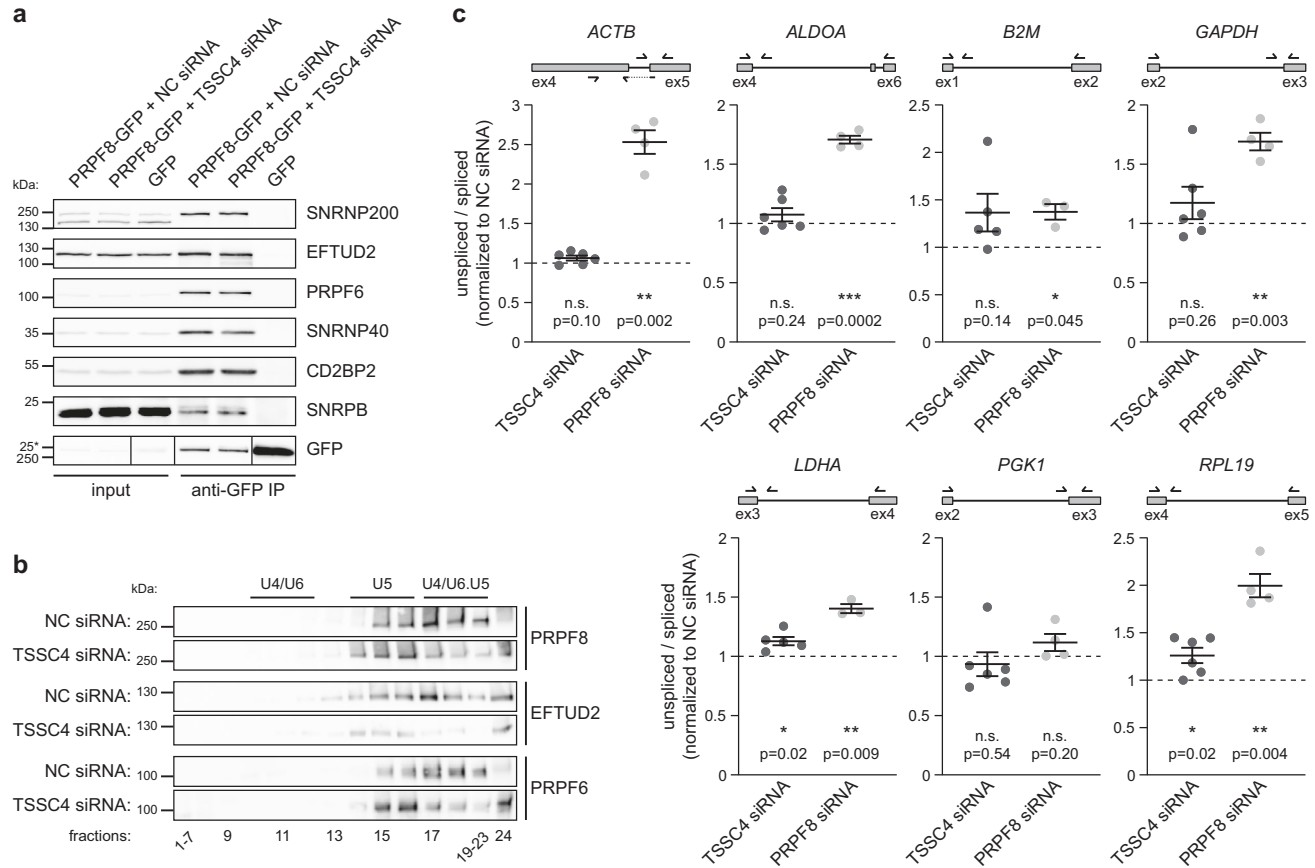

**Fig. 7 TSSC4 does not affect U5 snRNP composition but promotes tri-snRNP assembly. a** TSSC4 downregulation does not change the U5 snRNP protein composition. Co-immunoprecipitation of U5-specific proteins with PRPF8-GFP was assayed after TSSC4 knockdown by western blotting. The positions of molecular weight markers are shown, the marker labeled by asterisk applies only to the GFP sample in the bottom combined GFP panel. **b** Sedimentation of U5-specific proteins is altered after TSSC4 knockdown and all three U5 markers migrate in lighter fractions than in control extracts. The positions of molecular weight markers are shown. **a**, **b** The experiments were performed three times with similar results. Uncropped and unprocessed scans are provided in the Source Data file. **c** Splicing efficiency is not globally reduced after TSSC4 downregulation. Total RNA was isolated, reverse transcribed, and analyzed by qPCR. The knockdown of PRPF8 served as a positive control. Mean and SEM from at least three biologically independent experiments are shown (in each experiment reverse transcription was performed in two technical replicates and the average value used for calculations). Statistical significance was determined by the paired two-tailed T test, P values are shown. Values for graphs are provided in the Source Data file. See also Supplementary Fig. 4 for knockdown efficiency.

interactions between TSSC4 and U5 snRNP subunits: (1) Hom2 and Hom3 interacts with U5 snRNP where Hom2 primarily mediates association with the mature U5 snRNP and Hom3 with PRPF8/EFTUD2. (2) With SNRNP200 where the interaction platform is still unclear but it likely involves multiple regions. The tethering assay (Fig. 4) shows that TSSC4 binding to PRPF8, SNRNP200, and EFTUP2 can occur already in the cytoplasm, which suggests that TSSC4 interaction with RHC components happens primarily during their biogenesis. In addition, TSSC4 nuclear localization and binding to mature U5 snRNP and the U5/PRPF19 complex points to a function of TSSC4 in later stages of U5 maturation and post-splicing recycling.

A particle containing U5 snRNA, Sm, and RHC proteins emerges as an important intermediate in U5 snRNP metabolism where de novo biogenesis and recycling pathways merge. In both cases, additional U5-specific proteins (PRPF6, DDX23, CD2BP2, and TXNL4A) need to be added before U5 snRNP can associate with the U4/U6 di-snRNP. We hypothesize that TSSC4 acts in this step to support an optimal U5 snRNP structure compatible with tri-snRNP formation. In yeast, Aar2p co-purifies with the mature U5 snRNP and it has been suggested that Aar2p functions in U5 snRNP recycling after splicing[49]. In humans, AAR2 is

primarily cytoplasmic and co-precipitates only with a subset of U5 proteins and chaperones involved in RHC biogenesis[12]. TSSC4 might have evolved to replace the nuclear function of AAR2 in higher eukaryotes. Consistently, we did not identify TSSC4 homologs in unicellular eukaryotes and the first homolog was found in *Nematostella vectensis* (Supplementary Fig. 2c). Taking our data together, we propose that TSSC4 is a U5-specific chaperone that interacts with freshly assembled U5 snRNPs as well as with U5 particles recycled after splicing and promotes U5 snRNP interaction with the di-snRNP and formation of the splicing competent U4/U6•U5 tri-snRNP (Fig. 8c).

## Methods

**Cell culture.** HeLa cells, stable cell lines derived from HeLa cells, and mouse NIH-3T3 cells were cultured in high-glucose (4.5 g/l) Dulbecco's modified Eagle's medium (DMEM, Sigma-Aldrich) supplemented with 10% fetal bovine serum (FBS, Gibco) and 1% penicillin/streptomycin (Gibco). HeLa cells stably expressing PRPF8-GFP and SNRNP200-GFP from bacterial artificial chromosomes were kindly provided by Ina Poser (MPI-CBG, Dresden, Germany)[50] and characterized previously in ref. [47].

**Plasmids and transfection.** TSSC4 sequence was amplified from HeLa total RNA by reverse transcription followed by PCR using specific primers listed in

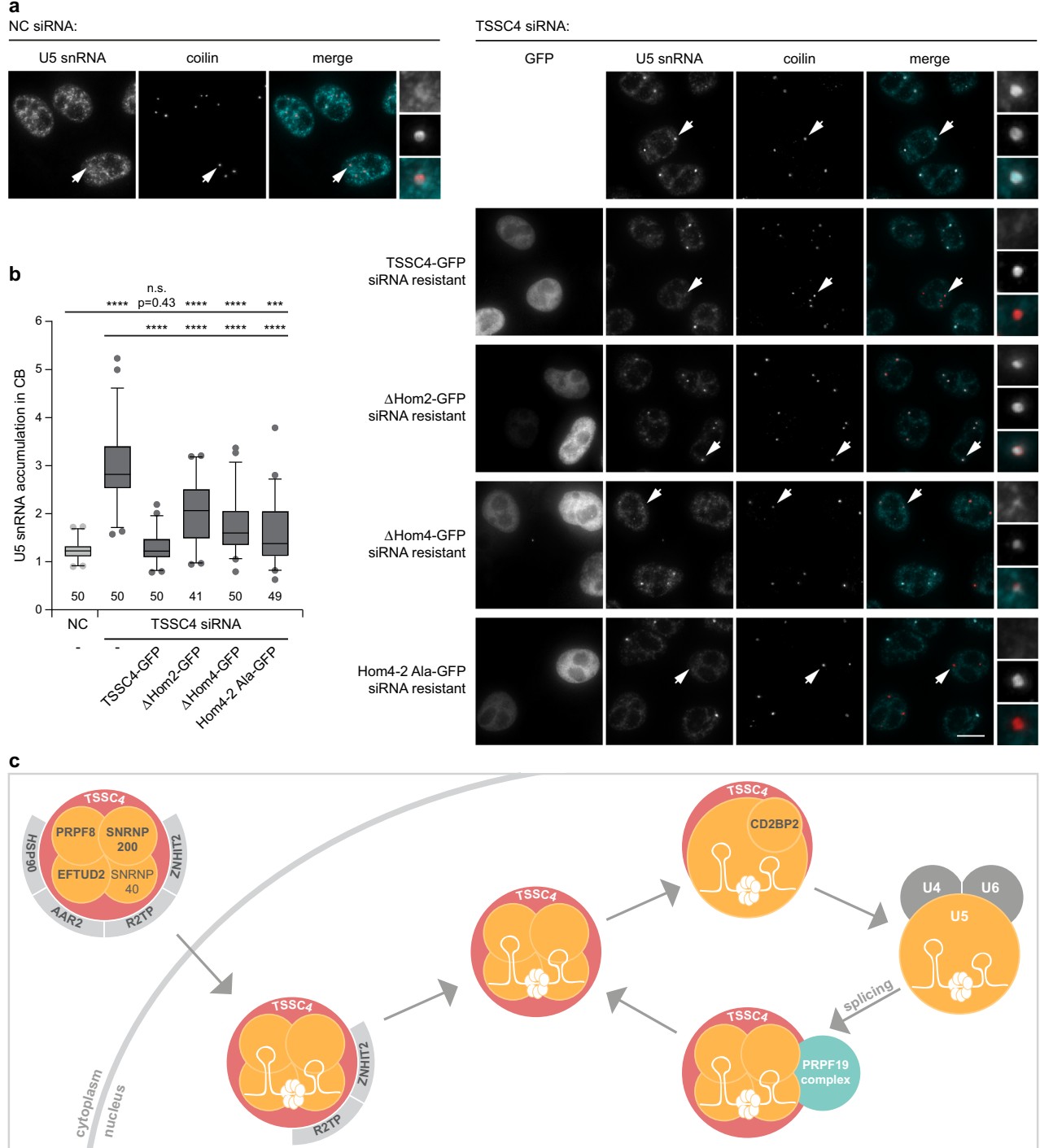

**Fig. 8 Hom2 and Hom4 regions are important for TSSC4 function in tri-snRNP formation. a** TSSC4 was downregulated by siRNA followed by expression of siRNA-resistant WT TSSC4-GFP, ΔHom2-GFP, ΔHom4-GFP, and Hom4-2_Ala-GFP constructs. Tri-snRNP formation was assayed by U5 snRNA localization (turquoise) in Cajal bodies visualized by immunodetection of their marker coilin (red). U5 snRNA localization in Cajal bodies is increased when tri-snRNP assembly is inhibited. Cajal bodies marked by arrows are enlarged five times in insets. Scale bar represents 10 μm. **b** Quantification of U5 snRNA Cajal body accumulation. In situ hybridization signal in the Cajal body was normalized to the average signal in the nucleoplasm. In total, 41–50 cells were assayed (see bottom of the diagram for the precise number of assayed cells) and results are shown as a box blot (center line—median; box limits—1st and 3rd quartiles; whiskers—5th and 95th percentile). Two lowest and two highest values are depicted as separate dots in each box plot. Statistical significance was determined by the two-tailed *T* test. *** indicates $P \leq 0.001$; **** indicates $P \leq 0.0001$. Original values for box plots are provided in the Source Data file. See also Supplementary Fig. 5 for correlation between TSSC4-GFP expression and U5 snRNA Cajal body accumulation. **c** A model of TSSC4 function in U5 snRNP metabolism.

Supplementary Table 2 and inserted into pEGFP-N1 (Clontech) and FLAG-C3 (pEGFP-C3 backbone where the GFP sequence was replaced with Triple FLAG) using BglII and SacII restriction sites. TSSC4 deletion mutants (Δ51–100, Δ101–150, Δ151–200, Δ201–250, Δ251–300, ΔHom2, ΔHom4, ΔHom2ΔHom3) and substitution mutants (Hom1-1_Ala, Hom1-2_Ala, Hom2-1_Ala, Hom2-2_Ala, Hom2-3_Ala, Hom4-1_Ala, Hom4-2_Ala, Hom4-3_Ala) were prepared by site-directed mutagenesis (for primer sequences see Supplementary Table 2). TSSC4 fragments (Hom2, Hom2Hom3, Hom4, Hom3Hom4) were amplified using specific primers (Supplementary Table 2) and inserted into pEGFP-N1 by XhoI/ KpnI restriction sites. All fragments were designed with ATG as a first codon. SiRNA-resistant TSSC4 constructs were prepared by mutating siRNA target site CGGTGG TGCTGAAGTGGAA (nt 123–141) to TGGCGGCGCGCAGAGGTCGAG using site-directed mutagenesis (Supplementary Table 2), the original amino acid sequence was preserved in siRNA-resistant constructs.

EFTUD2 was cloned into pEGFP-C3 vector using BamHI/XhoI restriction sites, CD2BP2 into pEGFP-N1 using BamHI/EcoRI sites, and PRPF6 into pFLAG-CMV-4 using HindIII/EcoRI sites. SNRPD1-pEGFP-C1 plasmid was kindly provided by A. Lamond (University of Dundee, UK). For the P-body tethering assay, TSSC4 was cloned into DDX6-dsRed-pcDNA5 FRT plasmid using the Gateway system (Invitrogen). All constructs were verified by DNA sequencing.

Plasmids were transiently transfected into the cells with Lipofectamine 3000 Transfection Reagent (Thermo Fisher Scientific) according to the manufacturer's instructions and cells analyzed 24 h post transfection.

**RNAi**. SiRNAs were transfected using the Oligofectamine Transfection Reagent (Thermo Fisher Scientific) according to the manufacturer's protocol. A single transfection was performed at a final concentration of 50 nM siRNA, and the cells were then incubated for 72 h. Alternatively, cells were transfected twice with 20 nM siRNA (final concentration) after 48 h. Both approaches resulted in comparable knockdown efficiency. Pre-annealed siRNA duplexes were obtained from Ambion: TSSC4 (s19605) 5′-CGGUGGUGCUGAAGUGGAAtt-3′, PRPF8 5′-CCU-GUAUGCCUGACCGUUUtt-3′[7], and negative control no. 5 siRNA.

**Antibodies**. For indirect fluorescent immunostaining, the following primary antibodies were used: anti-coilin (H-300, Santa Cruz Biotechnology; used 1:500), anti-FLAG (M2, Sigma-Aldrich; used 1:500), anti-PRPF19 (G-7, Santa Cruz Biotechnology; used 1:250) and anti-GFP (B-2, Santa Cruz Biotechnology; used 1:500), together with secondary antibodies conjugated with Alexa-488, Alexa-555 or Alexa-647 (Thermo Fisher Scientific; all used 1:750). Immunoprecipitation was performed using antibodies against GFP (provided by D. Drechsel, MPI-CBG, Dresden, Germany; used 6 μg per sample) and FLAG (M2 or F7425, Sigma-Aldrich; used 2 μg per sample). Following primary antibodies were used for western blotting: anti-TSSC4 (Proteintech; used 1:1000), anti-PRPF8 (E-5, Santa Cruz Biotechnology; used 1:250), anti-SNRNP200 (HPA029321, Sigma-Aldrich; used 1:500), anti-EFTUD2 (provided by R. Lührmann, MPI, Göttingen, Germany; used 1:1000), anti-PRPF6 (B-1, Santa Cruz Biotechnology; used 1:500), anti-SNRNP40 (PA5-55331, Thermo Fisher Scientific; used 1:3000), anti-CD2BP2 (PA5-59603, Thermo Fisher Scientific; used 1:6000), anti-DDX23 (PA5-34843, Thermo Fisher Scientific; used 1:1000), anti-TXNL4A (B-11, Santa Cruz Biotechnology; used 1:500), anti-PRPF19 (G-7, Santa Cruz Biotechnology; used 1:500), anti-CDC5L (D-11, Santa Cruz Biotechnology; used 1:500), anti-PLRG1 (E-12, Santa Cruz Biotechnology; used 1:100), anti-BCAS2 (F-5, Santa Cruz Biotechnology; used 1:250), AAR2 (SAB1104418, Sigma-Aldrich; used 1:500), anti-PRPF31 (ab188577, Abcam; used 1:4000), anti-SART3 (ab176822, Abcam; used 1:3000), anti-PRPF4 (provided by R. Lührmann, MPI, Göttingen, Germany; used 1:500), anti-SNRPA1 (ab128937, Abcam; used 1:4000), anti-GFP (B-2, Santa Cruz Biotechnology; used 1:500), anti-FLAG (M2, Sigma-Aldrich; used 1:1000), anti-FLAG (F7425, Sigma-Aldrich; used 1:400), anti-GAPDH (ab9484, Abcam; used 1:1000) and anti-Sm (Y12) produced from a hybridoma cell line (a gift from K. Neugebauer, Yale University, New Haven, USA) at the Antibody facility (Institute of Molecular Genetics of the Czech Academy of Sciences; used 1:100). Secondary antibodies conjugated with horseradish peroxidase (Jackson ImmunoResearch Laboratories) were used (1:10,000).

**Immunofluorescence, fluorescence in situ hybridization, and image acquisition**. Cells grown on coverslips were fixed with 4% paraformaldehyde/PIPES for 10 min at room temperature (RT), permeabilized with 0.2 or 0.5% Triton X-100 for 10 or 5 min, respectively, at RT and stained with primary antibodies diluted in phosphate buffered saline (PBS) for 1 h. After a PBS wash, the coverslips were stained with secondary antibodies for 1 h and then either mounted or further processed to achieve snRNA visualization. snRNAs were visualized by in situ hybridization using fluorescent DNA probes tagged on their 5′ end by Alexa-488 or Cy3. Probe sequences are listed in Supplementary Table 2. Cells were fixed again in 4% PFA/PIPES for 10 min and treated sequentially with 0.1 M glycine/0.2 M Tris-HCl (pH 7.4) for 10 min and 50% formamide/2× SSC for at least 10 min. The probe at a final concentration of 2 μM was hybridized in 50% formamide/10% dextran sulfate/1% BSA/2× SSC for 1 h at 37 °C. The final washing steps were performed as follows: 50% formamide/2x SSC for 20 min at 37 °C, 2× SSC for 20 min at 37 °C, and 1× SSC for 20 min at RT. The in situ hybridization protocol substantially

quenched GFP fluorescence. We, therefore, enhanced GFP visualization by indirect immunostaining with anti-GFP antibodies. All coverslips were mounted in Fluoromount-G (Southern Biotech) containing 4′,6-diamidino-2-phenylindole (DAPI).

Images were acquired using DeltaVision microscope system (Applied Precision Ltd.) coupled to the Olympus IX70 microscope equipped with a 60×/1.42 NA oil immersion objective, a CoolSNAP HQ2 camera (Photometrics; Princeton Instruments), and the acquisition software SoftWoRx (Applied Precision Ltd.). Stacks of 10–20 z sections with 200-nm z steps were taken per sample and subjected to mathematical deconvolution using SoftWoRx software. Maximum intensity projections of deconvolved pictures are presented. Fluorescent signal in Cajal bodies (CBs), P-bodies, and nucleoplasm was analyzed using Fiji software[51]. snRNA and protein accumulation in CB was calculated as [(mean intensity in CB)/ (mean intensity in nucleoplasm)]. Protein accumulation in P-bodies was calculated as [((mean intensity in P-body) – (mean intensity of background))/(mean intensity in nucleoplasm)]. Between 40 and 100 cells were analyzed for CB accumulation (cell numbers per sample are specified in individual figures) and 50 P-bodies were analyzed for P-body tethering assay.

**Heterokaryon assay**. HeLa cells were co-transfected with TSSC4-GFP and empty pmCherry-C3 plasmids. Alternatively, HeLa stably expressing PRPF8-GFP were transfected with pmCherry-C3 only. All HeLa were passaged 24 h post transfection, and $7 \times 10^5$ cells were seeded on coverslips together with $7 \times 10^5$ mouse NIH-3T3 cells and grown overnight.

The protocol for heterokaryon preparation was adapted from ref. [52]. Cells were treated with 50 μg/ml cycloheximide (CHX) or 0.5 μg/ml puromycin (Puro) for 2 h and then with 100 μg/ml CHX or 1 μg/ml Puro for 30 min to block protein synthesis. After a PBS wash, cells were fused by incubation in 50% PEG 4000 (Thermo Fisher Scientific) for 2 min. Fused cells were washed with PBS, incubated in DMEM supplemented with 100 μg/ml CHX or 1 μg/ml Puro for 3 h, and fixed with 4% paraformaldehyde/PIPES. Coverslips were mounted in Fluoromount-G medium containing DAPI and imaged as described above. The presence of mCherry protein in mouse NIH-3T3 nuclei served for the identification of heterokaryons.

**Immunoprecipitation**. Cells grown on a 10-cm Petri dish were harvested in ice-cold PBS, resuspended in NET2 buffer (50 mM Tris-HCl, pH 7.4, 150 mM NaCl, 0.05% Nonidet P-40) supplemented with Protease Inhibitor Cocktail Set III (Millipore) and RNasin ribonuclease inhibitor (Promega), and sonicated on ice bath by $2 \times 45$ pulses (0.5 s at 40% of maximum energy). Cell lysates were centrifuged at 20,000×g, 4 °C for 10 min, and then the supernatants were pre-cleaned by incubation with Protein G PLUS agarose beads (Santa Cruz Biotechnology) for 1 h at 4 °C and finally incubated overnight at 4 °C with anti-GFP or anti-FLAG antibodies pre-bound to agarose beads. Three percent of the lysate was put aside before the immunoprecipitation step and used as an input sample.

After the immunoprecipitation, beads were washed with NET2 buffer and proteins and/or RNA isolated. For protein analysis, the beads were directly mixed with SDS sample buffer (0.25 M Tris-HCl pH 6.8, 20% glycerol, 4% SDS, 2% β-mercaptoethanol, 0.02% bromphenol blue), and proteins resolved on a 10% polyacrylamide gel and detected by western blotting. RNA was extracted using phenol/chloroform, dissolved in urea sample buffer (20 mM Tris-HCl pH 8.0, 8 M urea, 0.2% xylene blue), and then resolved on a 7 M urea denaturing polyacrylamide gel and silver stained.

**Northern blot**. RNA was resolved on 7 M urea denaturing polyacrylamide gel and transferred by capillary to cationized Zeta-Probe nylon membrane (Bio-Rad). The transfer was performed for 16 h in 20× SSC buffer and RNA was then UV cross-linked to the membrane by 120 mJ/cm². Hybridization with a radioactively labeled probe was done according to ref. [53]. The DNA oligonucleotide probe complementary to U5 snRNA (Supplementary Table 2 and ref. [53]) was terminally labeled with 1 μl of [γ-³²P]ATP (3000 Ci/mmol, 10 mCi/ml) using T4 polynucleotide kinase, purified on Micro-Spin G25 columns (GE Healthcare) and denatured at 95 °C for 5 min. For pre-hybridization, the membrane was incubated in Church buffer (0.25 M sodium phosphate buffer pH 7.2, 1 mM EDTA, 1% BSA, 7% SDS) for 2 h at 65 °C. The probe was then added to the Church buffer and hybridized on the membrane at 65 °C overnight. After washing with Wash buffer-1 (2× SSC, 0.1% SDS) for 2× 10 min at 50 °C and Wash buffer-2 (0.2× SSC, 0.1% SDS) for 2× 10 min at 50 °C, the membrane was exposed to Storage Phosphor Screen and scanned on Amersham Typhoon 9500 (GE Healthcare).

**Glycerol-gradient ultracentrifugation**. TSSC4-FLAG co-precipitated complexes were eluted from the beads by incubation with $2 \times 25$ μg of triple FLAG peptide (Thermo Fisher Scientific) for 8 h and overnight at 4 °C. Nuclear extracts were prepared using NE-PER nuclear and cytoplasmic extraction reagents (Thermo Fisher Scientific) according to the manufacturer's instructions. Samples were diluted in gradient buffer (20 mM HEPES/KOH pH 8.0, 150 mM KCl, 1.5 mM MgCl₂) supplemented with Protease Inhibitor Cocktail Set III (Millipore), 0.5 mM PMSF, and 0.5 mM DTT and loaded on a linear 10–30% glycerol gradient. Complexes were fractionated by centrifugation at 130,000×g (32,000 rpm) using

SW-40 rotor (Beckman Coulter) for 17 h at 4 °C. Individual fractions (500 µl each, 24 fractions in total) were collected and from every fraction, RNA and proteins were isolated using TRIzol Reagent (Invitrogen) according to the manufacturer's protocol.

**SILAC labeling and proteomic analysis**. For SILAC experiments, HeLa Flp-In clones expressing GFP and FLAG-tagged TSSC4 were grown for 15 days in iso-topically labeled media (CIL/Eurisotop), to ensure complete incorporation of semi-heavy label L-lysine-$^2$HCl ($^2$H4, 96–98%)/L-arginine-HCl ($^{13}$C6, 99%) (percentages represent the isotopic purity of the labeled amino acids). Parental HeLa cells that did not express the tagged protein were used in control condition and were grown in a regular culture medium containing light (non-labeled) L-lysine-HCl/L-arginine-HCl. For each condition, $100 \times 10^6$ cells were trypsinized and cryogrinded before being resuspended in the lysis buffer (20 mM HEPES, pH 7.9, 100 mM KCl, 2 mM EDTA, 10% glycerol, 0.1% NP-40, 0.5% Triton X-100, protease inhibitor cocktail (cOmplete, Roche). Extracts were then incubated for 20 min at 4 °C, clarified by centrifugation for 10 min at 20,000×$g$, and finally pre-cleared by incubation with Protein G Sepharose beads (GE Healthcare) for 1 h at 4 °C. For SILAC-IP done with GFP tag, extracts were incubated with 50 µL of GFP-Trap beads (Chromotek) while for SILAC-IP with FLAG tag, agarose beads coated with M2 antibodies were used (Sigma-Aldrich). After incubating 1.5 h at 4 °C, beads were washed five times with lysis buffer, and the two isotopic conditions (light for control IP and medium for IP with tagged TSSC4) were finally pooled. Bound proteins were eluted by adding 1% SDS to the beads and boiling for 10 min. The eluates were reduced with DTT (10 mM, 2 min, 95 °C) and alkylated with iodoacetamide (50 mM, 30 min). Proteins were separated by SDS/PAGE and stained with colloidal blue. Each lane was cut in eight bands, and each band was gel-digested with trypsin (Promega V5280) in 100 mM triethylammonium bicarbonate. Digestion peptides were dried and resuspended in 0.1% formic acid/2% acetonitrile solution before being analyzed on a LTQ Velos Pro Orbitrap Elite mass spectrometer coupled to an Ultimate 3000 (Thermo Fisher Scientific) nanoflow chromatography. Desalting and pre-concentration of samples were performed on-line on a Pepmap precolumn (0.3 × 10 mm, Thermo Fisher Scientific) in buffer A (2% acetonitrile, 0.1% formic acid). A gradient consisting of 2–40% buffer B (B = 99.9% acetonitrile with 0.1% formic acid; 3–33 min) and 40–80% buffer B (33–34 min) was used to separate peptides at 300 nL/min from a Pepmap capillary reversed-phase column (0.075 × 150 mm, Thermo Fisher Scientific). Mass spectra were acquired using a top 20 collision-induced dissociation data-dependent acquisition method. The Orbitrap was programmed to perform a FT 400–1400 Th mass scan (60,000 resolution) with the top 20 ions in intensity selected for collision-induced dissociation data-dependent acquisition MS/MS in the LTQ. FT spectra were internally calibrated using a single lock mass (445.1200 Th). Target ion numbers were 500,000 for FT full scan on the Orbitrap and 10,000 MSn on the LTQ. Data were acquired using the Xcalibur software v2.2. Protein identification and quantitation were performed using the MaxQuant software v1.5.2.8 (http://www.maxquant.org/)[54]. Database for identification was Human Reference Proteome set (reviewed + isoforms) downloaded on May 25, 2017 from Uniprot.org. Proteins were identified with at least 2 matching peptides including at least one unique or Razor peptide (FDR < 1%, decoy approach). Significance calculations were performed with the Perseus v1.4.2 software.

**Splicing efficiency analysis**. Cells were harvested in ice-cold PBS, resuspended in NET2 buffer supplemented with Protease Inhibitor Cocktail Set III and RNasin ribonuclease inhibitor, and cell lysates were prepared as described above. Eight percent of the lysate was put aside, mixed with SDS sample buffer, and used for knockdown efficiency control by western blot. Total RNA was isolated from the rest of the cell lysate using TRIzol Reagent (Invitrogen) according to the manufacturer's protocol. The RNA pellet was resuspended in nuclease-free water, treated with Turbo DNase (Thermo Fisher Scientific) and again precipitated. Reverse transcription was performed with SuperScript III (Thermo Fisher Scientific) and random hexamers, and 1 or 5 µg of the total RNA was used per reaction.

cDNA was diluted 1:10 and analyzed by quantitative PCR using LightCycler 480 (Roche) and SYBR Green I reaction mix (Roche). Primer sequences are provided in Supplementary Table 2. Splicing efficiency (ratio pre-mRNA/mRNA) was calculated as $[2^{\text{Ct(mRNA)} - \text{Ct(pre-mRNA)}}]$. For every experiment, two samples with different amounts of total RNA used (1 and 5 µg) were analyzed and always the average of both values is presented.

**Yeast two-hybrid assay**. Plasmids pACTII and pAS2ΔΔ were introduced into haploid Saccharomyces cerevisiae strains (Y187 and CG1945, respectively). Strains were crossed and grown on YEPD and then plated on double- (–Leu–Trp) and triple-selective media (–Leu–Trp–His). Growth was assessed visually after 3 days at 30 °C. The strength of interactions was evaluated by comparing the number of clones growing on –Leu–Trp (selection of diploids) and –Leu–Trp–His plates (selection for interaction). Growth was sometimes performed with a gradual concentration of 3-amino-1,2,4-Triazol (3-AT) which is a competitive inhibitor of the HIS3 gene product.

## Data availability

The data that support this study are available from the corresponding author upon reasonable request. The mass spectrometry proteomics data generated in this study have been deposited to the ProteomeXchange Consortium via PRIDE[55] and are available with the identifier PXD024929. For TSSC4 sequence alignment, the following data were obtained from Uniprot: TSSC4 Homo sapiens—Q9Y5U2; TSSC4 Bos taurus—Q1LZD3; TSSC4 Mus musculus—Q9JHE7; TSSC4 Gallus gallus—Q5ZJS5; TSSC4 Danio rerio—Q0P4A6; TSSC4 Xenopus laevis—A0A1L8GD85; TSSC4 Drosophila melanogaster—Q9VT74; TSSC4 Nematostella vectensis—A7RVI2. The database used for identification of proteins after SILAC-IP was Human Reference Proteome set (reviewed + isoforms) downloaded from Uniprot.org (proteome ID UP000005640). The original microscopy images can be provided upon request. Source data are provided with this paper.

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

## Acknowledgements
We thank David Drechsler, Ina Poser, and Angus Lamond for providing us with reagents. This work was supported by the Czech Science Foundation (18-10035S and 21-04132S), the Agence Nationale de la Recherche [ANR-16-CE11-0032-04] and Institut national du Cancer [INCa PLBio 2016-161], the institutional funding (RVO68378050 and RVO68378050-KAV-NPUI), and the Ligue Nationale Contre le Cancer. The microscopy images were acquired at the Light Microscopy Core Facility, Institute of Molecular Genetics in Prague, Czech Republic supported by MEYS (LM2015062, CZ.02.1.01/0.0/0.0/16_013/0001775) and OPPK (CZ.2.16/3.1.00/21547). Mass spectrometry experiments were carried out using facilities of the Functional Proteomic Platform of Montpellier (FPP). Klara Klimesova, Jitka Vojackova, and Nenad Radivojevic were students of Faculty of Science, Charles University in Prague.

## Author contributions
K.K. performed all the experiments except SILAC-IP shown (Fig. 1a), snRNA, and proteins localization after TSSC4 knockdown (Figs. 5–6) and glycerol gradient after TSSC4 knockdown (Fig. 7b). J.V. made the initial observation regarding the TSSC4 function and contributed by snRNA and protein localization after TSSC4 knockdown (Figs. 5–6), glycerol-gradient ultracentrifugation after TSSC4 knockdown (Fig. 7b) and immunoprecipitation of TSSC4 shown at (Supplementary Figs. 1a and 2b). N.R. created selected TSSC4-GFP constructs (Fig. 3f) and performed initial IPs with them. F.V., E.B., and C.V. performed and analyzed TSSC4 SILAC-IP (Fig. 1a) and C.V. performed yeast two-hybrid assay (Supplementary Table 1). D.S. conceived the project and wrote the manuscript.

## Competing interests
The authors declare no competing interests.
