## [Peer Review File · Nature Communications]

REVIEWER COMMENTS

Reviewer #1 (Remarks to the Author):

Besides requiring specific pathways for de novo biosynthesis, spliceosomal snRNPs undergo major restructuring during splicing and thus need to be reassembled after splicing. De novo biosynthesis and recycling of spliceosomal snRNPs involves chaperones and trafficking between cellular and sub-nuclear compartments, and is still poorly understood on the molecular level. Klimesova et al. present data suggesting that TSSC4 associates early with U5 snRNP during biogenesis, promotes U4/U6-U5 snRNP assembly and may be involved in U5/PRPF19 complex recycling after splicing. They demonstrate association of TSSC4 with U5 snRNP and PRPF19 complex components via immunoprecipitation followed by mass spectrometry and Western blot. Using systematic deletion variants, they delineate TSSC4 regions important for certain interactions. They use a P-body recruitment assay to provide evidence that TSSC4 and certain U5 snRNP components can associate in the cytoplasm. Based on immunofluorescence microscopy and FISH studies monitoring localization of spliceosomal components to Cajal bodies upon TSSC4 knockdown (and rescue experiments with TSSC4 and deletion variants), and based on redistribution of U5 snRNP components in glycerol gradients upon TSSC4 knockdown they suggest that TSSC4 is required for U4/U6-U5 tri-snRNP assembly.

The manuscript deals with an important, underexplored aspect of spliceosome function, the de novo biogenesis and recycling of its subunits. It is generally well written. The work seems to be technically sound and involves laborious assays with many systematically designed TSSC4 constructs.

Specific points:

1. As a general point the paper remains largely descriptive and does not provide major new insights into the molecular mechanisms of snRNP biogenesis or recycling. How TSSC4 carries out its presumed functions remains in the dark. The data provide no firm evidence, which of the detected interactions are direct and which can occur at the same time/in the same complex.
2. It is not clear from the description if the proteomics analyses were performed once or several times. How did the authors decide on the cutoff for what to consider TSSC4 interactors? Differences in enrichment are quite high for some highlighted proteins in FLAG-IP compared to GFP-IP.
3. Deletion studies with TSSC4 assume that the observed effects can be explained by deleted regions representing interaction regions for binding partners. This will only hold true if TSSC4 is intrinsically unfolded. For a folded protein there may be severe indirect effects due to misfolding of larger regions. The authors only state that the protein does not contain recognizable domains, but is it unstructured? What is the evidence?
4. As presented, the differential Cajal body accumulations of proteins (Fig. 5, 7A) are not convincing. The authors should quantify, as apparently done for U5 in Fig. 7B.
5. Figure 1C: What is the thick band above U5? Also RNA detection by, e.g., Northern blotting would be more convincing, especially for FLAG-TSSCA IP in Figure 2A.

6. Figure 2A: The authors state that one complex sedimented in fractions 13-15, in the figure it rather looks like fractions 14-16. Also see above point: It is not possible to decide based on the data shown that TSSC4 preferentially interacts with U5 and U5/PRPF19 compared to tri-snRNP. It rather looks like all U5-containing complexes can be pulled by TSSC4 (similar apparent relative intensity distribution for U5 in both panels).
7. Figure 3B: Delta-201-250 seems to show reduced binding of SNRNP200 and PRPF8 - could this be the reason for reduced binding of PRPF19?
8. The authors show that all TSSC4 constructs locate to the nucleus. In light of this, their P-body recruitment assay in the cytoplasm is somewhat confusing. Is there enough TSSC4 in the cytoplasm for early association with U5 proteins? How does nuclear-cytoplasmic distribution look for known snRNP assembly chaperones?
9. The splicing defects presented for TSSC4 knockdown in Figure 6 are not convincing. Additional targets should be tested if the authors want to document a splicing defect.
10. It should be mentioned that TSSC4 interactions with some of the proteins found as interactors here have previously been observed in high-throughput studies (e.g. Huttlin et al., Nature 2017).

Reviewer #2 (Remarks to the Author):

The U5 snRNP is an important element in the canonical splicing process and is part of the splicing competent U4/U6•U5 tri-snRNP. The assembly of the U5 snRNP has been well examined over the years and the many factors involved have been identified and characterized, as well as the subcellular regions in which parts of the maturation process take place such as the nuclear Cajal body. The Stanek group has made significant contributions to the understanding of snRNP biogenesis in Cajal bodies. Here, they identify a protein called TSSC4 to be involved in U5 snRNP biogenesis and its subsequent recycling, as well as the interactions occurring with U5 snRNP components. The interaction of TSSC4 with U5 snRNP proteins was first identified by IPs and proteomics, and further confirmed by a series of pulldowns including the pulldown of U5 snRNA, and the finding that TSSC4 interactions are RNA independent. Use of a glycerol gradient demonstrated that TSSC4 partners in two distinguishable complexes: a mono-U5 snRNP particle and the PRPF19 complex. Using truncations of TSSC4, the region of aa 51-100 was identified as being critical for the interactions with U5 proteins and snRNA, while aa 201-250 were the interaction domain with PRPF19. Further detailed analysis with a whole additional large set of domain regions plus some mutations revealed which domains and the amino acids within interact with the different U5 complexes.

In order to determine whether the interaction of TSSC4 with its partners takes place already in the cytoplasm prior to import into the nucleus, the authors use a tethering system that localizes the target protein into cytoplasmic P bodies and then examines if any of the protein partners are drawn to this structure as well. Indeed, only a subset of U5 proteins were found associated with the anchor TSSC4, and these were proteins of the RHC complex. To look at later stages of biogenesis, the levels of TSSC4 were knocked down and the levels of U snRNA in Cajal bodies were determined, and altogether experiments confirmed that TSSC4 is important for U4/U6•U5 tri-snRNP formation. Further experiments determine the domains in TSSC4 that are important for this.

This is a very nice study. The data are well presented and clean, notably the biochemistry and complex western blots are quite exquisite such that the effort put into this study is noticeable. In my opinion, the study delineates a novel set of interactions occurring in the U5 snRNP at different stages of its life cycle, and the detailed analysis performed here is convincing as to the different roles TSSC4 plays. Altogether, this study deserves to get published and I only suggest some minor issues to be corrected.

Minor comments:

* Page 8: "TSSC4 downregulation increased Cajal body accumulation of U4, U5 and U6 snRNAs but not U2 snRNA, which indicated specific defects in U5 snRNP biogenesis." Although this can be seen in figure 5 to some extent for a few Cajal bodies, a proper quantification would be more suitable for this figure (as in Fig. 7B).

* Page 9 – fig. 6C: here this quantification of splicing efficiency but where are the actual data.

* Fig. S3 – quantify so we know levels of KD

Reviewer #3 (Remarks to the Author):

In this work, Klimesova et al have investigated the role of a novel protein, TSSC4, in its role to affect U5snRNP biogenesis by interacting with U5-associated proteins and then its ability to influence tri-snRNP assembly. In this work, the authors have used standard approaches for these types of analyses and, in general, the many experiments presented have a very high technical quality. In general, the manuscript does a good job in showing how new potentially splicing-important cellular factors can be identified even after several decades of research in this area. The major drawback of the work, in the opinion of this reviewer, is that a major effort has been dedicated at mapping precisely the interactions between TSSC4 and other spliceosomal factors at the expense of providing strong evidence that the interaction between TSSC4 and U5snRNP has a clear functional meaning. For this reason, the clarifications and additions that should be provided by the authors according to this reviewer are as follows:

1) As the authors point out towards the end of the Introduction, the connection between TSSC4 and U5 biogenesis was hypothesized following co-precipitation with U5 chaperones and include ECD, TSSC4, and NCDN. Considering that ECD has already been studied, they have concentrated on TSSC4. However, NCDN has not been looked at and it would be interesting to know why out of these two remaining proteins the authors decide to focus on TSSC4. Is there a particular reason?. For example, considering the number of U5-related factors reported in Figure 1A are they not surprised that NCDN was absent from this list?.

2) In Figure S2A the authors show that all deletion mutants of TSSC4 localize to the cell nucleus (only one nucleus is shown that is not really ideal). However, they do not show the level of expression of these proteins in HeLA cells. As these proteins are then extensively used in IP studie (Figure 3) it would be important to show that they are all expressed at the same level simply by performing a western blot against GFP.

3) In Figure 3 the authors perform experiments with an impressive number of mutants. However, they are all deletion mutants that might introduce some artefactual results due to changes in the distance between different domains. Have the authors considered expressing just the 51-100 aa and 201-250 sequence to see whether they are able to interact with the proteins shoes binding was disrupted by their deletion?.

4) Figure 4 needs some form of quantification. The very high resolution used in this figure, although it is adequate to show co-localization, does not really allow to understand in an unbiased manner the specific differences between the TSSC4-DDX6-dsRed and the five U5 specific proteins co-expressed in these cells.

5) In Figure 5A, if the authors express a si-resistant TSSC4 can they rescue the accumulation of U4, U5, and U6 in Cajal bodies?. There is always the possibility, in fact, that the siRNA might have some aspecific targets that could induce the observed accumulation. At the very least, as this is a key results of the manuscript, the authors should show that the same effect can be obtained using a second siRNA against TSSC4 that has a sequence different from the first.

6) The changes in splicing efficiency in Figure 6C for TSSC4 are really very modest. It is therefore rather surprising that the authors have tested only two housekeeping genes. Surely, there are many more genes that could be tested in order to make this conclusion more convincing.

7) Taken together, the authors work suggest that TSSC4 interacts with the U5snRNP abut not with the U4/U6.U5 trisnRNP but in Figure 2A the signal for the Flag-TSSC4 IP seems to overlap considerably with the first U4/U6.U5 fraction. Are they really sure to rule out some residual TSSC4 presence in the tri-snRNP?.

Point-by-point response to reviewers' comments

Reviewer #1

The manuscript deals with an important, underexplored aspect of spliceosome function, the de novo biogenesis and recycling of its subunits. It is generally well written. The work seems to be technically sound and involves laborious assays with many systematically designed TSSC4 constructs.

Thank you!

Specific points:

1. As a general point the paper remains largely descriptive and does not provide major new insights into the molecular mechanisms of snRNP biogenesis or recycling. How TSSC4 carries out its presumed functions remains in the dark. The data provide no firm evidence, which of the detected interactions are direct and which can occur at the same time/in the same complex.

To address the reviewers concern we performed yeast 2-hybrid assay and tested TSSC4 interaction with all eight U5-specific proteins and the PRPF19 protein. Two large proteins PRPF8 and SNRNP200 were divided into six and five fragments, respectively (see a scheme below) and each of these fragments was tested for interaction with TSSC4 independently. Unfortunately, we did not detect any specific interaction between TSSC4 and any of the tested proteins. These data suggest that TSSC4 associates with pre-assembled U5 complexes and/or makes multiple weak contacts with several U5 components that are below detection limit of the yeast 2-hybrid assay. We added a short paragraph about the assay and its results (p. 8, middle).

2. It is not clear from the description if the proteomics analyses were performed once or several times. How did the authors decide on the cutoff for what to consider TSSC4 interactors? Differences in enrichment are quite high for some highlighted proteins in FLAG-IP compared to GFP-IP.

The pull down was performed once for TSSC4-GFP and once for TSSC4-FLAG. Protein identification and quantitation were performed using the MaxQuant software v1.5.2.8 (<http://www.maxquant.org/>). Database for identification was Human Reference Proteome set (reviewed + isoforms) downloaded on May 25th 2017 from Uniprot.org. Proteins were identified with at least 2 matching peptides including at least 1 unique or Razor peptide

(FDR<1%, decoy approach). Significance calculation were performed with the Perseus v1.4.2 software. Significance A test is an outlier significance score (pvalue) for log protein ratios of protein intensities. It determines which values are significant outliers relative to a certain population considering a normal distribution. In our case, majority of the quantified proteins ratios population correspond to the background of purification and we are trying to determine which proteins display enrichment ratios significantly different from the general population (outliers). Significance B is the same as Significance A, but intensity-dependent. Due to mass spectrometry limitations, for highly abundant proteins the statistical spread of unregulated (unenriched) proteins is much more focused than for low abundance ones. To capture this effect, significance B is calculated only on proteins subsets obtained by intensity binning. Then, B significance is the probability of obtaining a log-ratio of at least this magnitude under the null hypothesis that the distribution of log-ratios has normal upper and lower tails. For details see Cox and Mann (2008) Nat. Biotech. 26, 1367-72. For the presentation of SILAC IP data in Fig. 1A, we highlighted only those proteins that co-purified with M/L ratio higher than 1.2 in FLAG IP and were detected in GFP IP at the same time. All other proteins detected in the analysis were presented as grey dots. Complete list of co-precipitated proteins is in Supplementary Table 1.

3. Deletion studies with TSSC4 assume that the observed effects can be explained by deleted regions representing interaction regions for binding partners. This will only hold true if TSSC4 is intrinsically unfolded. For a folded protein there may be severe indirect effects due to misfolding of larger regions. The authors only state that the protein does not contain recognizable domains, but is it unstructured? What is the evidence?

We analyzed TSSC4 sequence by MetaDisorderMD2 program to predict folded and disordered regions. This analysis suggested that TSSC4 is mostly unstructured protein and only conserved regions showed partially ordered structure. These data are now included in Supplementary figure S2D. To further address this concern, we cloned and expressed conserved regions tagged with GFP and analyzed their association with U5 and PRPF19 complexes. These data showed that the central part of TSSC4 comprising conserved region Hom2 and Hom3 interacts with U5-specific proteins and weakly also with PRPF19 complex components. Co-precipitation of PRPF19 complex might indicate that Hom2-3 construct is able to associate with U5/PRPF19 particle. These data suggest that conserved domains Hom2-3 represent necessary and sufficient binding platform for U5 snRNP. The new results are included in Fig. 3G

4. As presented, the differential Cajal body accumulations of proteins (Fig. 5, 7A) are not convincing. The authors should quantify, as apparently done for U5 in Fig. 7B.

The quantification of snRNAs and U5 components in Cajal bodies has been done and is now included in Figures 5 and 6.

5. Figure 1C: What is the thick band above U5? Also RNA detection by, e.g., Northern blotting would be more convincing, especially for FLAG-TSSCA IP in Figure 2A.

We performed a new TSSC4 immunoprecipitation and detected co-precipitated RNAs with silver stain (Fig. 1C, top panel) and specifically U5 snRNA with Northern blotting (Fig. 1C, bottom panel). Similarly, we performed a new immunoprecipitation of TSSC4-FLAG followed

by analysis of coprecipitated particles by gradient ultracentrifugation but the signal was unfortunately weak and we were not able to unambiguously distinguish co-precipitated complexes from background signal. However, we believe that immunoprecipitation of U5 snRNA identified by Northern blotting (Fig. 1C) as well as U5-specific proteins (Fig. 1 and 2) give us strong affirmations that TSSC4 interacts with U5 snRNP including its RNA component U5 snRNA. The thick band above U5 in silver-stained gels is 5S rRNA and we marked it in the revised version to clearly point that out.

6. Figure 2A: The authors state that one complex sedimented in fractions 13-15, in the figure it rather looks like fractions 14-16. Also see above point: It is not possible to decide based on the data shown that TSSC4 preferentially interacts with U5 and U5/PRPF19 compared to tri-snRNP. It rather looks like all U5-containing complexes can be pulled by TSSC4 (similar apparent relative intensity distribution for U5 in both panels).

We looked over our glycerol gradients and corrected the sedimentation of mono U5 snRNP in fractions 14-16 as suggested by the reviewer, Nevertheless, we believe that U5-containing complexes coprecipitating with TSSC4 differ from U5-containing complexes found in nuclear extracts. Namely, U5-containing particles sedimenting in fractions 17-18 in nuclear extracts are underrepresented in TSSC4 pull down. These are the fractions where tri-snRNP sediments and these data are fully consistent with classical immunoprecipitation where we do not detect U4/U6-specific Prpf31 in TSSC4 pulldowns indicating that TSSC4 does not associate with the tri-snRNP (Fig. 1B).

7. Figure 3B: Delta-201-250 seems to show reduced binding of SNRNP200 and PRPF8 - could this be the reason for reduced binding of PRPF19?

The reviewer is right that Delta 201-250 construct co-precipitates slightly lower amount of PRPF8 and SNRNP200. However, this reduction is rather marginal with respect to almost complete loss of PRPF19 and other components of the PRPF19 complex. We would rather incline to opposite explanation and suggest that removal of 201-250 region weakens association with the post-spliceosomal U5/PRPF19 particle and that's why we might observe reduced co-precipitation of U5-specific proteins.

8. The authors show that all TSSC4 constructs locate to the nucleus. In light of this, their P-body recruitment assay in the cytoplasm is somewhat confusing. Is there enough TSSC4 in the cytoplasm for early association with U5 proteins? How does nuclear-cytoplasmic distribution look for known snRNP assembly chaperones?

The reviewer is right that there is a discrepancy between localization of TSSC4 in the nucleus and observed interaction with U5 proteins in the cytoplasm. To address this issue, we tested whether TSSC4 cycle between the nucleus and the cytoplasm by a heterokaryon assay (new Fig. S3). We expressed TSSC4-GFP in HeLa cells, fused them with mouse NIH-3T3 cells, inhibited translation using two different inhibitors and monitored appearance of TSSC4-GFP in mouse nuclei. HeLa and NIH-3T3 cells have very distinct chromatin pattern and are easily distinguishable. Our results show that TSSC4 shuttle between nucleus and the cytoplasm. The other U5-specific chaperon AAR2 is mainly cytoplasmic (e.g. Malinova et al. 2017, J. Cell Biol.; <https://doi.org/10.1083/jcb.201701165>), the R2TP complex is found mainly in the nucleus and nucleolus. We point to this localization in last paragraph of

Discussion when we speculate about TSSC4 function.

9. The splicing defects presented for TSSC4 knockdown in Figure 6 are not convincing. Additional targets should be tested if the authors want to document a splicing defect. *We tested splicing of additional five genes (total seven genes assayed) after TSSC4 and PRPF8 knockdown. While PRPF8 knockdown significantly reduced splicing of 6 out of 7 tested genes, TSSC4 had a much milder effect and we observed partial inhibition of splicing for two genes only. These new data are now presented at Figure 7C.*

10. It should be mentioned that TSSC4 interactions with some of the proteins found as interactors here have previously been observed in high-throughput studies (e.g. Huttlin et al., Nature 2017).

We included the reference to Huttlin et al. 2017 that identified several TSSC4 interactors into the last paragraph of Introduction (p. 3) as suggested.

Reviewer #2

This is a very nice study. The data are well presented and clean, notably the biochemistry and complex western blots are quite exquisite such that the effort put into this study is noticeable. In my opinion, the study delineates a novel set of interactions occurring in the U5 snRNP at different stages of its life cycle, and the detailed analysis performed here is convincing as to the different roles TSSC4 plays. Altogether, this study deserves to get published and I only suggest some minor issues to be corrected.

Thank you for nice comments on our manuscript!

Minor comments:

* Page 8: "TSSC4 downregulation increased Cajal body accumulation of U4, U5 and U6 snRNAs but not U2 snRNA, which indicated specific defects in U5 snRNP biogenesis."

Although this can be seen in figure 5 to some extent for a few Cajal bodies, a proper quantification would be more suitable for this figure (as in Fig. 7B).

Quantification of U5 component accumulation in Cajal bodies (Figs. 5 and 6) and P-bodies (Fig. 4) was included as suggested by the reviewer.

* Page 9 – fig. 6C: here this quantification of splicing efficiency but where are the actual data.

To monitor splicing efficiency we employed RT-qPCR and primer pairs amplifying a region spanning exon-intron junctions to detect unspliced RNAs and over exon-exon junctions to detect spliced RNAs. Then the splicing efficiency was calculated as a ratio between unspliced and spliced RNAs (please, see material and Methods, p. 17 for detailed description) and normalized to cells treated with negative control siRNA. To clarify this point, we included schemes depicting primer localization on tested genes above the graphs (Fig. 7C).

* Fig. S3 – quantify so we know levels of KD

Knockdown efficiency of TSSC4 and PRPF8 has been quantified as suggested (Fig. S4B).

Reviewer #3

1) As the authors point out towards the end of the Introduction, the connection between TSSC4 and U5 biogenesis was hypothesized following co-precipitation with U5 chaperones and include ECD, TSSC4, and NCDN. Considering that ECD has already been studied, they have concentrated on TSSC4. However, NCDN has not been looked at and it would be interesting to know why out of these two remaining proteins the authors decide to focus on TSSC4. Is there a particular reason?.

In our previous work, we repeatedly detected TSSC4 in all pulldowns with U5-specific proteins PRPF8 and EFTUD2 as well as co-precipitating with U5 snRNP chaperons AAR2 and ZNHIT2 (Malinova et al. 2017, J. Cell Biol.; <https://doi.org/10.1083/jcb.201701165>). That's why we focused on this protein. We added an explanatory note into the last paragraph of Introduction (p. 3) to explain our choice.

For example, considering the number of U5-related factors reported in Figure 1A are they not surprised that NCDN was absent from this list?

NCDN was detected in the pulldown with TSSC4-FLAG but not in the pulldown with TSSC4-GFP (see Table S1) and therefore does not stand out in Fig. 1A, where we highlighted only proteins coprecipitating with both GFP and FLAG versions of TSSC4.

2) In Figure S2A the authors show that all deletion mutants of TSSC4 localize to the cell nucleus (only one nucleus is shown that is not really ideal). However, they do not show the level of expression of these proteins in HeLA cells. As these proteins are then extensively used in IP studies (Figure 3) it would be important to show that they are all expressed at the same level simply by performing a western blot against GFP.

Almost all TSSC4 deletion/substitution mutants are expressed at very similar levels as wild-type TSSC4-GFP. The only exception is a deletion mutant lacking Hom1 region and the constructs comprising Hom2 or Hom3-4 domains, which are expressed at reduced levels and we mention this fact in Results (p. 7, middle of first paragraph and p. 8 top). Expression of all TSSC4 constructs is shown on Fig. 3B, D, E and G, bottom left panel (GFP signal in inputs).

3) In Figure 3 the authors perform experiments with an impressive number of mutants. However, they are all deletion mutants that might introduce some artefactual results due to changes in the distance between different domains. Have the authors considered expressing just the 51-100 aa and 201-250 sequence to see whether they are able to interact with the proteins whose binding was disrupted by their deletion?

This is an excellent point. To address this issue, we expressed TSSC4 constructs comprising different conserved regions (and surrounding sequences) Hom2, Hom2-3, Hom3-4 and Hom4 (see Fig. 3F for schemes of these new constructs). Immunoprecipitation followed by detection of U5 and PRPF19 complex components showed that Hom2-3 construct is able to co-precipitate all tested U5 snRNP proteins. We also detected a weaker association with PRPF19 complex factors, which might indicate interaction of Hom2-3 construct with the U5/PRPF19 particle. These data are consistent with the deletion experiments where deletion of Hom2 and 3 domains significantly reduced interaction of

TSSC4 with U5 proteins (except SNRNP200). These new data are now shown at Fig. 3G and described in Results (p.7 bottom and p.8 top).

4) Figure 4 needs some form of quantification. The very high resolution used in this figure, although it is adequate to show co-localization, does not really allow to understand in an unbiased manner the specific differences between the TSSC4-DDX6-dsRed and the five U5 specific proteins co-expressed in these cells.

As requested, we quantified localization of U5 proteins in P-bodies and the data are now shown in Fig. 4B.

5) In Figure 5A, if the authors express a si-resistant TSSC4 can they rescue the accumulation of U4, U5, and U6 in Cajal bodies?. There is always the possibility, in fact, that the siRNA might have some aspecific targets that could induce the observed accumulation. At the very least, as this is a key results of the manuscript, the authors should show that the same effect can be obtained using a second siRNA against TSSC4 that has a sequence different from the first.

To show that U4, U5 and U6 accumulation in Cajal bodies is due to downregulation of TSSC4, we expressed siRNA-resistant TSSC4 and were able to revert the phenotype observed after TSSC4 knockdown (Fig. 8A and B).

6) The changes in splicing efficiency in Figure 6C for TSSC4 are really very modest. It is therefore rather surprising that the authors have tested only two housekeeping genes. Surely, there are many more genes that could be tested in order to make this conclusion more convincing.

We tested splicing of five additional genes (total seven genes assayed) after TSSC4 and PRPF8 knockdown. While PRPF8 knockdown significantly reduced splicing of 6 out of 7 tested genes, TSSC4 had a much milder effect and we observed partial inhibition of splicing for two genes only. These new data are now presented at Figure 7C. The data show that in contrast to knockdown of core splicing factor PRPF8, downregulation of TSSC4 has only modest effect on splicing efficiency.

7) Taken together, the authors work suggest that TSSC4 interacts with the U5snRNP abut not with the U4/U6.U5 trisnRNP but in Figure 2A the signal for the Flag-TSSC4 IP seems to overlap considerably with the first U4/U6.U5 fraction. Are they really sure to rule out some residual TSSC4 presence in the tri-snRNP?.

The reviewer is correct that we cannot fully rule out the possibility that some residual TSSC4 associates with the tri-snRNP. However, when we combine results from immunoprecipitation (where we do not detect any U4/U6-specific proteins Fig. 1A and B) with glycerol gradient ultracentrifugation where endogenous TSSC4 sediments in lower fractions than the tri-snRNP components (Fig. 2), the simplest explanation of these data is that the majority of TSSC4 does not interact with the tri-snRNP. However, we soften our statement about TSSC4 association with the tri-snRNP in Discussion (p. 11, top paragraph) to address reviewer's concern.

REVIEWERS' COMMENTS

Reviewer #1 (Remarks to the Author):

The authors have thoroughly addressed all points raised by this reviewer. In particular, they have added new data to the revised manuscript that strengthen their original conclusions.

Reviewer #3 (Remarks to the Author):

Authors have fully answered the queries raised by this reviewer and have made their case much stronger